# HORIZON IMAGINATION: EFFICIENT ON-POLICY ROLLOUT IN DIFFUSION WORLD MODELS

**Lior Cohen**[*]
Technion

**Ofir Nabati**
Technion

**Kaixin Wang**
Microsoft Research

**Navdeep Kumar**
Technion

**Shie Mannor**
Technion

## ABSTRACT

We study diffusion-based world models for reinforcement learning, which offer high generative fidelity but face critical efficiency challenges in control. Current methods either require heavyweight models at inference or rely on highly sequential imagination, both of which impose prohibitive computational costs. We propose Horizon Imagination (HI), an on-policy imagination process for discrete stochastic policies that denoises multiple future observations in parallel. HI incorporates a stabilization mechanism and a novel sampling schedule that decouples the denoising budget from the effective horizon over which denoising is applied while also supporting fractional steps-per-frame budgets (sub-step budgets). Experiments on Atari 100K and Craftium show that our approach maintains control performance with a sub-step budget of half the denoising steps (i.e., 0.5 denoising steps per frame) and achieves superior generation quality under varied schedules. Code is available at `https://github.com/leor-c/horizon-imagination`.

## 1 INTRODUCTION

World models (Ha & Schmidhuber, 2018) have emerged as a powerful paradigm for achieving sample-efficient reinforcement learning (RL). By learning a generative model of environment dynamics, world models enable agents to produce large quantities of simulated experience that can be used to train controllers, thereby reducing reliance on costly real environment interactions. Recently, diffusion world models have become particularly attractive, owing to their unmatched generative fidelity across modalities such as images and video (Esser et al., 2024; Blattmann et al., 2023).

Despite these advantages, current approaches face key limitations when applied to control. Practical deployment often requires controllers that are lightweight and highly computationally efficient, enabling real-time inference with low power consumption. In contrast, the prevailing trend in world models, particularly diffusion-based models, is toward ever-larger architectures (Agarwal et al., 2025; Parker-Holder et al., 2024; Ball et al., 2025; Assran et al., 2025). This poses a significant barrier to their direct use at test time, as generation becomes increasingly time- and compute-intensive. Furthermore, methods that rely on repeated interaction with the world model during controller inference are therefore impractical in many real-world settings.

Alternatively, approaches that train an independent controller via world model imagination (Alonso et al., 2024) encounter prohibitive computational overhead with diffusion-based models. Since each observation must be generated through a costly multi-step denoising process, and imagination requires sequentially interleaving policy decisions with world model predictions, the resulting imagination process is inherently sequential and computationally intensive.

In this work, we propose a more efficient approach for training discrete stochastic policies with diffusion world models. We introduce Horizon Imagination, an on-policy imagination process that denoises multiple future observations simultaneously, reducing the sequential burden of diffusion generation. We devise (i) a mechanism to mitigate policy-induced instability during horizon imagination, and (ii) a novel sampling schedule that disentangles the denoising budget from the schedule's decay horizon, enabling independent control over both parameters, supporting fractional steps-per-frame budgets (sub-step budgets), and yielding superior generation quality at higher budgets. Our

---

[*]Email: liorcohen5@campus.technion.ac.il

approach is training-agnostic and applies to any pre-trained world model with observation-level time conditioning.

We validate our approach on a subset of Atari 100K and Craftium environments, showing that the agent maintains control performance with only half the denoising budget. We further analyze world model generation quality under different configurations of the proposed Horizon schedule, spanning fully autoregressive to highly parallel regimes and a range of denoising budgets. Parallel generation consistently proves advantageous, achieving strong performance even under sub-step budgets.

## 2 RELATED WORK

**Large Diffusion World Models**   Diffusion-based methods have emerged as the state of the art in generative modeling, delivering unmatched fidelity across modalities such as images and video (Ho et al., 2020; Rombach et al., 2022; Ho et al., 2022). Building on this success, recent large-scale world models adopt diffusion backbones for their superior generative capabilities (Ball et al., 2025; Agarwal et al., 2025; Decart et al., 2024; Assran et al., 2025). While motivated by applications such as agent training, these works are limited to conditional video generation and do not address control.

**Concurrent Multi-step Generation**   The idea of generating multiple steps in parallel with diffusion world models has been explored in several recent works (Chen et al., 2024; Rigter et al., 2024; Ding et al., 2024; Jackson et al., 2024). In particular, Diffusion Forcing (Chen et al., 2024) introduces a multi-step simultaneous generation scheme together with a control framework in which the world model serves as both policy and planner, akin to model predictive control (García et al., 1989). Although effective for learning control, this approach is computationally intensive and often impractical for deployment. In contrast, our work addresses this limitation by enabling efficient training in imagination, allowing a lightweight policy to be extracted from the world model for deployment.

Rigter et al. (2024) and Jackson et al. (2024) proposed policy-guided imagination methods where during the denoising process the actions are updated in the direction of the score of the policy action distribution $\nabla_a \pi(a|s)$, as in Langevin dynamics (Song & Ermon, 2019) and classifier guidance (Dhariwal & Nichol, 2021). However, these approaches are limited to continuous action spaces.

Ding et al. (2024) proposed an off-policy multi-step generation approach for the offline RL setting. Importantly, none of the above approaches examined how world model generation quality is affected by sampling configurations, including sequential versus parallel generation and the influence of the generation budget.

**Diffusion World Model Agents for Control**   Alonso et al. (2024) and Yang et al. (2024) proposed diffusion-based world model methods that follow a traditional step-by-step sequential imagination. Both reported significant computational overhead. In particular, Alonso et al. (2024) provided a detailed runtime analysis, showing that sequential imagination dominates the overall cost.

## 3 PRELIMINARIES

**Reinforcement Learning Setup**   Consider a Partially Observable Markov Decision Process (POMDP) environment. Here, we consider a practical state-agnostic formulation, as the agent has no knowledge about the POMDP hidden state space. At each step $t = 1, 2, \ldots$, starting from some initial observation $\mathbf{o}_1$, the agent picks an action $\mathbf{a}_t$ and the environment evolves according to $\mathbf{o}_{t+1}, r_t, d_t \sim p(\mathbf{o}_{t+1}, r_t, d_t | \mathbf{o}_{\leq t}, \mathbf{a}_{\leq t})$, where $r_t \in \mathbb{R}, d_t \in \{0, 1\}$ are the reward and termination signals, respectively. This process repeats until a non-zero termination signal is observed. The agent's goal is to maximize its expected return $\mathbb{E}[\sum_{t=0}^{\infty} \gamma^t r_{t+1}]$.

**World Model Agents**   In reinforcement learning (RL), "world models" (Ha & Schmidhuber, 2018) denote a class of model-based methods where the controller is trained entirely on simulated rollouts generated by a learned dynamics model, whereas real environment interactions are used for training the model. World-model agents typically consist of four components: a representation model for encoding and decoding observations, a dynamics model, a controller, and a replay buffer.

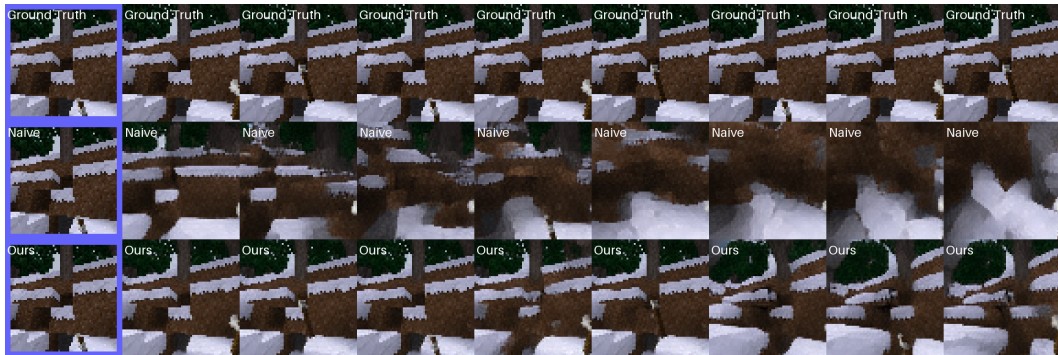

Figure 1: Example of generation instabilities observed in `Craftium/ChopTree-v0` under naive action sampling during horizon imagination. In contrast, our stable sampling method produces robust, high-quality generations. The first context frame is highlighted with a blue border.

World model agents are trained either in an online or an offline setting. Online training learns the components jointly from scratch, fitting the representation and dynamics to real interactions while optimizing the controller via model-generated rollouts, whereas offline training uses pre-collected data to train the components sequentially: representation, dynamics, and controller. Our method is applicable in both settings.

**Diffusion Framework**   In this work, we employ the 1-Rectified Flow (RF) framework (Liu et al., 2023) for its simplicity and robustness, though our method remains compatible with other diffusion variants. Formally, given samples from an unknown target distribution $p_{\text{data}}$, RF learns a time-dependent vector field $v_\theta$ parameterized by $\theta$ that transports samples $x^0 \sim p_{\text{prior}}$ from a chosen prior distribution $p_{\text{prior}}$ toward data samples $x^1 \sim p^1 \approx p_{\text{data}}$. The evolution is governed by

$$\mathrm{d}x^\tau = v_\theta(x^\tau, \tau)\,\mathrm{d}\tau, \quad \tau \in [0,1],$$

where superscripts denote denoising time. The vector field $v_\theta$ is trained to estimate $\mathbb{E}[x^1 - x^0 \mid x^\tau]$, with $x^\tau = \tau x^1 + (1-\tau)x^0$, by minimizing the regression objective

$$L(\theta) = \mathbb{E}_{x^0, x^1, \tau} \left\| v_\theta(x^\tau, \tau) - (x^1 - x^0) \right\|^2.$$

## 4   METHOD

Multi-step trajectory generation naturally alternates between producing an observation $\mathbf{o}_t$ and sampling an action $\mathbf{a}_t \sim \pi(\cdot \mid \mathbf{o}_{\leq t}, \mathbf{a}_{<t})$, as each step depends on the output of the previous one. However, since the generative process of diffusion methods involves multiple (costly) forward passes for generating each observation, the above sequential generation of a trajectory becomes highly sequential and expensive, both in time and in compute.

We propose a more efficient approach where the denoiser $v_\theta$ denoises multiple observations in parallel, conditioned on actions that are updated to track the policy as it co-evolves with the world model and incorporates newly available information. However, because action changes alter subsequent observations through different dynamics, they may in turn trigger further updates to both the policy and observations and destabilize the denoising process (Figure 1). Our approach therefore aims to minimize unnecessary action changes due to stochasticity, while ensuring that actions remain aligned with the evolving policy.

### 4.1   WORLD MODEL TRAINING

The world model is a learned generative model of the environment dynamics

$$p(\mathbf{z}_{t+1}, r_t, d_t \mid \mathbf{z}_{\leq t}, \mathbf{a}_{\leq t}),$$

operating over sequences of compact latent representations $\mathbf{z}_t$ of observations $\mathbf{o}_t$. These latents are produced by the encoder of the representation model, whose output is constrained to

$\mathbf{z}_t \in [-1, 1]^{d_\mathrm{C} \times d_\mathrm{H} \times d_\mathrm{W}}$ via a `tanh` activation. The world model comprises a denoiser $v_\theta$ for generating future observations and a separate, lightweight reward–termination predictor. This design cleanly separates a large, general-purpose dynamics module from small, task-specific predictors.

During training, $h$-step trajectory segments comprising observations $\mathbf{o}_{1:h}$, actions $\mathbf{a}_{1:h}$, rewards, and terminations are sampled from the replay buffer. The observations $\mathbf{o}_{1:h}$ are encoded into latent representations $\mathbf{z}_{1:h}$, serving as the clean targets $\mathbf{z}^1 \sim p_\mathrm{data}$. A uniform noise sample $\mathbf{z}^0 \sim p_\mathrm{prior} = \mathcal{U}([-1, 1])^{h \times d_\mathrm{C} \times d_\mathrm{H} \times d_\mathrm{W}}$ of matching shape is also drawn. Following Chen et al. (2024), an independent denoising time is sampled for each observation, i.e., $\tau \sim \mathcal{U}([0, 1])^h$. With probability 0.2, a clean prefix is provided by setting $\tau_{\leq \mathrm{k}} = 1$, where $\mathrm{k} \sim \mathcal{U}(\{1, \ldots, \lfloor 0.7h \rfloor\})$. We empirically found that it improves generation quality by better matching inference-time conditions, where the initial context is noise-free.

Given $\mathbf{z}^1 \sim p_\mathrm{data}$, $\mathbf{z}^0 \sim p_\mathrm{prior}$, and $\tau$, the denoiser $v_\theta$ is trained to approximate

$$\mathbb{E}_{\mathbf{z}^0, \mathbf{z}^1, \tau}[\mathbf{z}_t^1 - \mathbf{z}_t^0 \mid \mathbf{z}_1^{\tau_1}, \ldots, \mathbf{z}_t^{\tau_t}, \mathbf{a}_{<t}, \tau_{\leq t}], \quad \forall 1 \leq t \leq h,$$

by minimizing the rectified flow regression objective

$$L(\theta) = \frac{1}{h} \sum_{t=1}^h \mathbb{E}_{\mathbf{z}^0, \mathbf{z}^1, \tau} \| v_\theta(\mathbf{z}_1^{\tau_1}, \ldots, \mathbf{z}_t^{\tau_t}, \mathbf{a}_{<t}, \tau_{\leq t}) - (\mathbf{z}_t^1 - \mathbf{z}_t^0) \|^2, \tag{1}$$

where $\mathbf{z}_t^{\tau_t} = \tau_t \, \mathbf{z}_t^1 + (1 - \tau_t) \, \mathbf{z}_t^0$. All denoiser outputs are computed in parallel in a single forward pass. Each output corresponds to a particular observation at timestep $t$, and uses only inputs up to its timestep, preserving causality. Further details, including reward–termination modeling, are provided in Appendix A.1.2. Pseudocode is given in Appendix C.

## 4.2 Horizon Imagination (World Model Inference)

Denoising multiple observations in parallel during imagination implies denoising far-future observations before near-future ones are fully clean. Moreover, denoising such far-future observations requires access to future actions up to that timestep (see Eq. 1). These actions therefore must be conditioned on intermediate, still-noisy observations. We therefore query the policy before each denoising step, yielding an increasingly informed categorical action distribution for each timestep. Later, the policy is optimized across all noise levels based on its imagined interactions.

However, naively sampling from these distributions can lead to frequent action changes across denoising steps, particularly when the policy outputs high-entropy distributions, even when the policy itself remains fixed. To address this, we introduce a stable sampling method that reduces unnecessary action changes throughout the denoising process. For clarity, we present the method at the level of a single action at some arbitrary time $t$.

**Minimizing Action Changes Over the Denoising Process** Consider a set of $N$ discrete actions $\mathcal{A} = [N]$. At denoising time $\tau$, the policy induces a categorical distribution $\pi^\tau = \pi(\cdot | \mathbf{z}_{\leq t}^\tau, \mathbf{a}_{<t}) \in \Delta^{N-1}$ where $\Delta^{N-1}$ is the standard $(N-1)$-simplex. Inspired by inverse transform sampling, our method leverages a single multivariate uniform sample that is consistently mapped to an action under the evolving distributions $\pi^\tau$.

At the beginning of the denoising process, we draw two random objects: (i) a vector $\omega \sim \mathcal{U}([0, 1))^{N-1}$, and (ii) a random permutation $\rho : [N] \to [N]$ that fixes an order over the actions. At each denoising time $\tau$, the action is given by $\mathbf{a}^\tau = \mathbf{a}(\pi^\tau, \omega)$, ensuring that the same uniform sample $\omega$ yields consistent decisions across the evolving distributions $\pi^\tau$. The action $\mathbf{a}(\pi, \omega)$ is determined by scanning $\omega$'s coordinates:

$$\mathbf{a}(\pi, \omega) = \begin{cases} \rho(i) & \text{for the smallest } i \text{ with } \omega_i < \alpha_i(\pi), \\ \rho(N) & \text{if no such } i \text{ exists}, \end{cases} \tag{2}$$

where

$$\alpha_i(\pi) = \begin{cases} \frac{\pi_{\rho(i)}}{S_i(\pi)} & S_i(\pi) > 0 \\ \pi_{\rho(i)} & S_i(\pi) = 0 \end{cases}, \qquad S_i(\pi) = \sum_{j=i}^N \pi_{\rho(j)}.$$

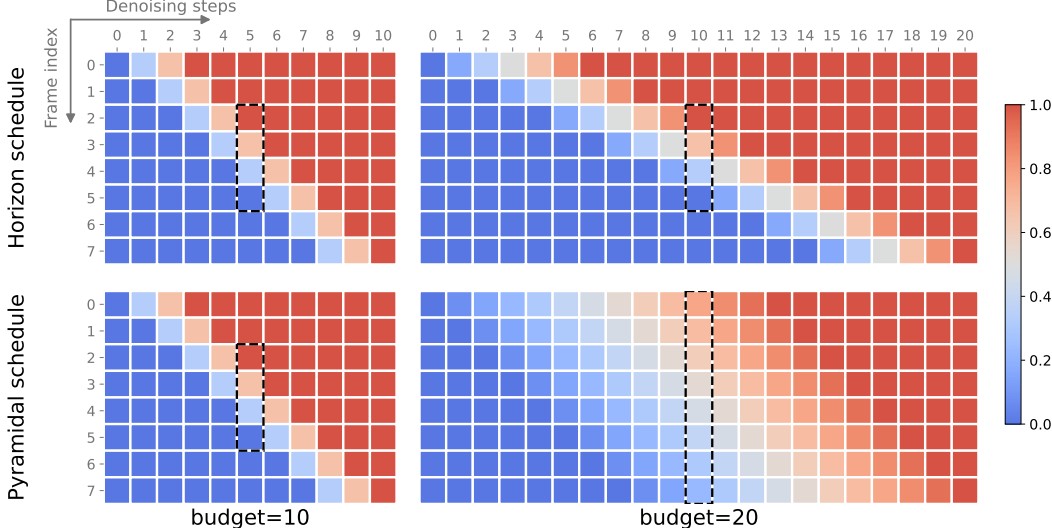

Figure 2: Comparison of the Pyramidal schedule (Chen et al., 2024) and the proposed Horizon schedule (transposed). Horizon fixes the decay horizon ($\nu = 3$), yielding consistent schedules across budgets, whereas in the Pyramidal schedule the decay horizon drifts with budget, as the two are entangled, leading to degraded generation quality at higher budgets.

Intuitively, $\alpha_i(\pi)$ is the conditional probability of choosing the $i$-th action in the order $\rho$, given that all earlier actions were skipped.

**Proposition 1.** *The sampling scheme* $\mathrm{a}(\cdot, \cdot)$ *satisfies the following properties:*

1. *For any distribution $\boldsymbol{p} \in \Delta^{N-1}$, the random variable $\mathrm{a}(\boldsymbol{p}, \omega)$ with $\omega \sim \mathcal{U}([0,1))^{N-1}$ is distributed according to $\boldsymbol{p}$, i.e., $\mathrm{a}(\boldsymbol{p}, \cdot) \sim \boldsymbol{p}$.*

2. *For any $\boldsymbol{p}, \boldsymbol{q} \in \Delta^{N-1}$ (e.g., $\boldsymbol{p} = \pi^{\tau_1}$ and $\boldsymbol{q} = \pi^{\tau_2}$ with $\tau_1 < \tau_2$), consider the event*
$$A = \{\omega \mid \mathrm{a}(\boldsymbol{p}, \omega) \neq \mathrm{a}(\boldsymbol{q}, \omega)\}.$$
*Let $\delta(\boldsymbol{p}, \boldsymbol{q}) = \frac{1}{2} \|\boldsymbol{p} - \boldsymbol{q}\|_1$ denote the total variation distance between $\boldsymbol{p}$ and $\boldsymbol{q}$. Then*
$$\delta(\boldsymbol{p}, \boldsymbol{q}) \leq \Pr(A) \leq \|\alpha(\boldsymbol{p}) - \alpha(\boldsymbol{q})\|_1.$$

We defer the proof to Appendix E. As an immediate corollary, if $p = q$ then $\Pr(A) = 0$, meaning that when the distribution does not vary between denoising steps, the actions necessarily remain unchanged. In contrast, naively sampling a fresh action at each step is likely to induce frequent action changes under sufficiently high-entropy policies.

**The Horizon Schedule** We propose a novel time schedule inspired by the pyramidal schedule of Chen et al. (2024), which denoises near future observations before far future ones. Unlike prior approaches, our schedule *disentangles* the rate at which denoising progresses across time, i.e., the *decay horizon*, from the *denoising budget*, i.e., how many denoising steps are performed in total. This design provides precise and independent control over both parameters.

Formally, let $\nu \in [1, h]$ specify the decay horizon, i.e., the number of steps over which the denoising schedule decays from 1 to 0, and let $B \in \mathbb{N}$ denote the total budget of denoising steps. We define the sampling schedule as a matrix $\boldsymbol{K} \in [0, 1]^{(B+1) \times h}$, where each of the first $B$ rows specifies the denoising times $\tau$ assigned to all $h$ observations at a given step. The last row ensures that $\tau_t = 1$ for all $1 \leq t \leq h$. To determine the values of the entries of $\boldsymbol{K}$, consider the lines
$$\kappa(t, b) = -\frac{1}{\nu}t + \frac{b}{B}\left(1 + \frac{h-1}{\nu}\right),$$
with slope $-\frac{1}{\nu}$, sequence time index $0 \leq t < h$, and denoising step index $0 \leq b \leq B$. The entries of $\boldsymbol{K}$ are given by
$$K_{i,j} = \mathrm{clamp}(\kappa(j-1, i-1), 0, 1). \tag{3}$$

---

**Algorithm 1** Horizon Imagination

---

**Require:** World model denoiser $v_\theta$, policy $\pi$, generation horizon $h$, denoising budget $B$, decay horizon $\nu$, reward and termination models $r_\phi, d_\phi$, optional context segment $(\mathbf{z}_{\leq k}, \mathbf{a}_{<k})$ with $k \geq 0$

1: Compute the Horizon schedule $\boldsymbol{K}$ given $h, B, \nu$          ▷ (Eq. 3)
2: Sample $\mathbf{z}_{k+1}^0, \ldots, \mathbf{z}_{k+h}^0 \sim p_{\text{prior}}$
3: Sample $\omega_t, \rho_t$ for every $k + 1 \leq t \leq k + h - 1$
4: **if** $k > 0$ **then**
5:      Sample first action $a_k \sim \pi(\cdot \,|\, \mathbf{z}_{\leq k}, \mathbf{a}_{<k})$
6: **end if**
7: **for** each denoising iteration $b$ from 1 to $B$ **do**
8:      Set $\tau \leftarrow \boldsymbol{K}_{b,:}$ and $\mathrm{d}\tau \leftarrow \boldsymbol{K}_{b+1,:} - \boldsymbol{K}_{b,:}$
9:      Denote $\mathbf{z}_{\leq k+t}^\tau = (\mathbf{z}_{\leq k}^1, \mathbf{z}_{k+1}^{\tau_1}, \ldots, \mathbf{z}_{k+t}^{\tau_t})$
10:      **for** $1 \leq t \leq h - 1$ **do**
11:          Compute action distributions $\pi_{k+t}^\tau \leftarrow \pi(\cdot \,|\, \mathbf{z}_{\leq k+t}^\tau, \mathbf{a}_{<k+t}^\tau, \tau_{\leq t})$
12:          Generate actions $a_{k+t}^{\tau_t} = a\left(\pi_{k+t}^\tau, \omega_t\right)$          ▷ (Eq. 2)
13:      **end for**
14:      Perform a denoising step: $\mathbf{z}_{k+t}^{\tau_t + \mathrm{d}\tau_t} \leftarrow \mathbf{z}_{k+t}^{\tau_t} + v_\theta(\mathbf{z}_{\leq k+t}^\tau, \mathbf{a}_{<k+t}^\tau, \tau_{\leq t})\,\mathrm{d}\tau_t$
     for all $1 \leq t \leq h$ with $\mathrm{d}\tau_t > 0$          ▷ (single forward pass)
15: **end for**
16: Compute rewards and terminations $\bar{r}_{\leq k+h}, d_{\leq k+h}$ with $r_\phi, d_\phi$
17: **return** imagined trajectory $\mathbf{z}_{\leq k+h}^1, \mathbf{a}_{<k+h}, \bar{r}_{\leq k+h}, d_{\leq k+h}$

---

Importantly, the proposed schedule allows any combination of $B$ and $\nu$, and in particular $B < h$. Thus, while standard auto-regressive generation requires $B \geq h$, and is limited to multiples of $h$, our schedule can use arbitrary positive integer budget values, including fractional steps-per-frame budgets (sub-step budgets).

### 4.3 ACTOR-CRITIC TRAINING

In the previous sections, we described how the interaction between the policy and the world model unfolds. These interactions generate the data used to optimize the controller. Unlike standard imagination, our parallel generation requires the policy to produce outputs at every denoising step, from pure noise to fully denoised samples. Consequently, the policy must be trained across all noise levels to ensure meaningful outputs.

We build on an actor-critic method introduced in prior works (Cohen et al., 2025; Hafner et al., 2025). The actor and critic are implemented as separate networks with identical backbones and distinct linear heads for their respective outputs. The critic operates only on fully denoised inputs, providing bootstrapped value estimates for the actor. The actor is trained using the standard REINFORCE objective (Sutton et al., 1999) applied at every trajectory step before termination, leveraging interactions generated across all noise levels. Formally, conditioned on a context $\mathbf{z}_{\leq k}^1, \mathbf{a}_{<k}$ of $k$ clean frames, the policy objective for trajectory time $k + t$ and denoising time $\tau_t$ is given by

$$A_{k+t} \log \pi(a_{k+t}^{\tau_t} | \mathbf{z}_{\leq k}^1, \mathbf{z}_{k+1}^{\tau_1}, \ldots, \mathbf{z}_{k+t}^{\tau_t}, \mathbf{a}_{<k+t}, \tau) + \eta \mathcal{H}(\pi_{k+t}^{\tau_t}),$$

where $A_t$ is the advantage at step $t$ and $\mathcal{H}$ is the policy entropy. To ensure balanced policy updates over denoising times, we restrict updates to steps in which the denoising time of the next observation increases. Full actor–critic details are given in Appendix A.1.3.

## 5 EXPERIMENTS

### 5.1 EMPIRICAL ANALYSIS OF ACTION CONSISTENCY

We conduct two controlled experiments to evaluate the proposed stable action sampling mechanism.

In the first experiment (Figure 3a), we examine the empirical distribution of average action changes between random source and target distributions $\boldsymbol{p}, \boldsymbol{q}$ using our method. We sample $10^4$ pairs of

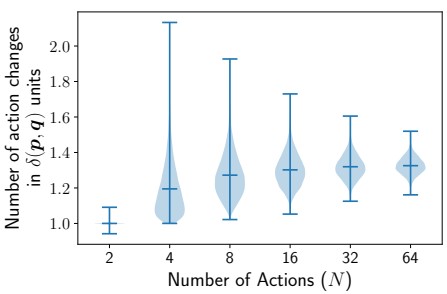 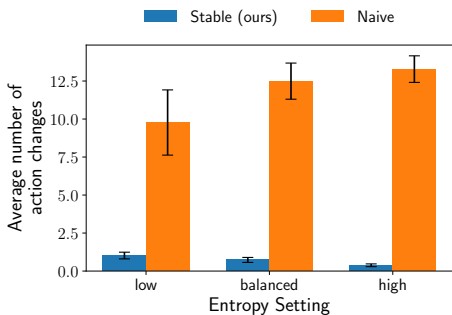

(a) Distributions of the average number of action changes, in $\delta(\boldsymbol{p}, \boldsymbol{q})$ units, for various $N$ values. Minimum, mean, and maximum are indicated.

(b) Comparing the number of action changes between naive sampling and the proposed method (mean $\pm$ std).

Figure 3: Empirical study of the average number of action changes under various settings.

categorical distributions from a uniform Dirichlet prior with $N$ actions. For each pair, we draw $10^6$ $(\omega, \rho)$ samples, compute the induced actions, and record the average number of changes. This is repeated for various $N$ values. Empirically, our method performs very close to the tight theoretical lower bound, the total variation $\delta(p, q)$, across all $N$ values, demonstrating near-optimal performance.

In the second experiment (Figure 3b), we compare naive sampling with the proposed stable mechanism in a simulated denoising process. We interpolate between a source and target categorical distribution with $N = 10$ actions across 8 steps, then fix the target distribution for 8 additional steps. Both source and target are sampled from a Dirichlet distribution under three entropy settings: low $(0.2, \dots, 0.2)$, uniform, and high $(5, \dots, 5)$. For each entropy setting, we simulate the process for 1,000 source-target pairs and estimate the average number of action changes across all 16 steps over $10^6$ simulations per pair. Our results show that while the proposed method yields at most one action change on average across all 16 steps, naive sampling incurs changes in most steps, altering more than half overall. For high-entropy distributions, the number of action changes decreases under our method but increases with naive sampling, consistent with our theoretical results.

## 5.2 Control Performance

To study whether Horizon Imagination enables a more efficient training, we benchmark our world model agent in an online RL setting. The agent's training process involves a repeated cycle of four stages: data collection through environment interactions, representation learning, world model learning, and actor-critic learning in imagination. For comparison, we consider three baselines that differ only in their imagination configuration: a standard autoregressive baseline with 32 denoising steps in total (one step per observation), denoted $(\nu = 1, B = 32)$, and two variants with decay horizon $\nu = 4$ and budgets of $B = 16$ and $B = 32$.

### 5.2.1 Setup

**Agent Implementation** For latent representations, we employ the continuous autoencoder image tokenizer of Agarwal et al. (2025), and apply a final `tanh` activation to the encoder output to constrain values to $[-1, 1]$. The denoiser network $v_\theta$ is implemented as an action-conditioned causal Diffusion Transformer (DiT) (Peebles & Xie, 2023) architecture. This model operates on sequences of latent observations, where action and denoising time conditioning are carried via the adaptive layer normalization (AdaLN) layers of the DiT architecture. Rewards and termination signals are modeled using a lightweight recurrent neural network (RNN) with a compact convolutional neural network (CNN) feature extractor and two output heads, operating on encoded trajectories. The same RNN architecture is used for both the actor and critic networks. For simplicity, we omit action conditioning in the actor, critic, and reward-termination models, as in all benchmarks considered the observations alone provide sufficient information. The agent uses fixed hyperparameters across

all experiments and comprises a tokenizer (22.5M), world model (67M), and actor-critic (7.5M), totaling 97M parameters. We defer the full implementation details to Appendix A.

**Benchmarks** We evaluate on four Atari100K (Kaiser et al., 2020) games and four Craftium (Malagón et al., 2025) games, both benchmarks involving visual inputs and discrete actions. All environments were run for 100K interaction steps, except `Craftium/SmallRoom-v0`, which was capped at 30K due to its lower difficulty. The well-established Atari benchmark serves as a solid control evaluation suite and is visually simpler. In practice, we found that even a single denoising step can yield satisfactory generation quality in Atari. By contrast, Craftium presents richer and more complex observations, providing a more challenging test for the world model.

**Training Time and Hardware** All experiments were conducted on NVIDIA A100 GPUs, except for `Craftium/SmallRoom-v0`, which was run on RTX 5090, RTX 4090, and A40 GPUs. On Atari, training with a denoising budget of $B = 32$ required approximately 27 hours per run, while reducing the budget to $B = 16$ shortened training to about 19 hours. Because the tokenizer and world model training stages are unaffected by this parameter, the overall speed-up is smaller than a full $2\times$. On Craftium, runs take longer due to slower environment interactions.

### 5.2.2 RESULTS

As shown in Figure 4, both baselines with decay horizon $\nu = 4$ maintain the performance of the autoregressive baseline across all environments, achieving comparable results at reduced cost. Notably, our approach sustains full performance with a sub-step budget of $B = 16$, requiring only half the denoising steps of the autoregressive baseline.

Moreover, our results suggest that a single denoising step per observation generally suffices, with baselines at $B = 32$ performing comparably across environments. In contrast, in `Craftium/-ChopTree-v0`, the most visually complex environment, the $(\nu = 4, B = 32)$ baseline achieves superior performance, suggesting improved effectiveness under the $B = 32$ budget.

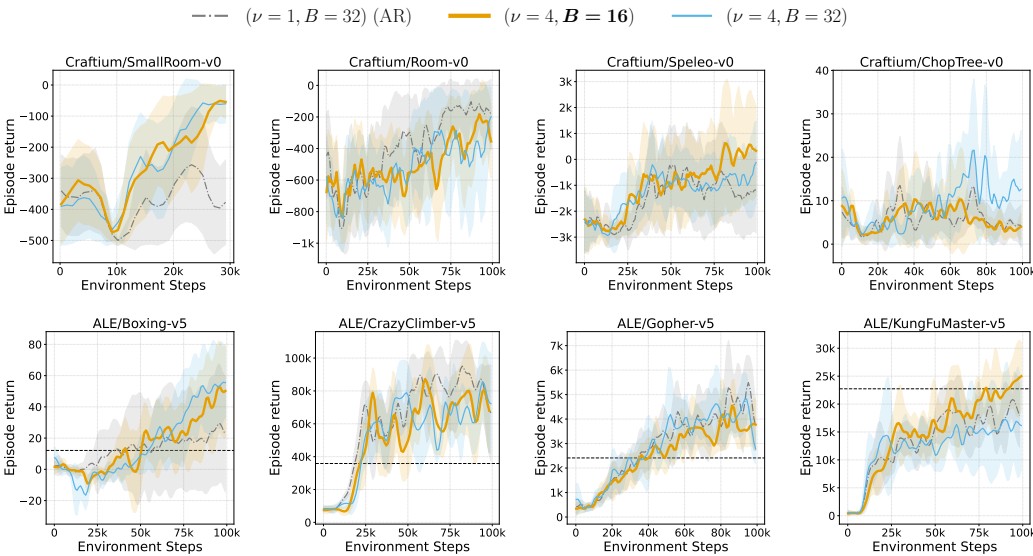

Figure 4: Actor-Critic Performance. Average episodic return curves of key baselines during training. Each baseline is evaluated over 5 seeds. Curves show the mean and standard deviation, smoothed by a moving average (window size 15). A dashed horizontal line denotes Atari human-level performance.

### 5.2.3 ABLATION STUDY: STABLE ACTION SAMPLING

To assess the influence of the proposed stable action sampling mechanism on control performance, we perform an ablation study by reverting to naive action sampling. In this baseline, a fresh action is independently drawn from the policy distribution before each denoising step.

The results in Figure 5 show that replacing the stable action sampling mechanism with naive sampling leads to a substantial drop in control performance (returns). Notably, while the naive baseline is consistently weaker, we observe a particularly prominent collapse on Atari Boxing and Gopher. These findings further highlight the importance of preventing unnecessary action fluctuations during the denoising process.

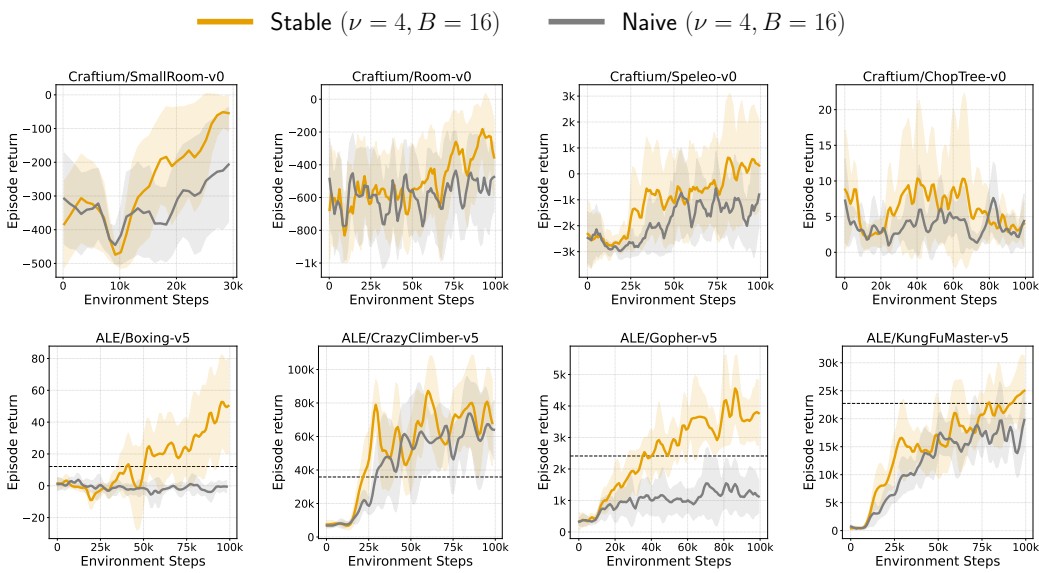

Figure 5: Actor–critic performance comparison between the proposed stable action sampling method and the naive baseline. Each baseline is evaluated over 5 seeds. Curves show the mean and standard deviation, smoothed by a moving average (window size 15). A dashed horizontal line denotes Atari human-level performance.

### 5.3 PARALLEL VS. SEQUENTIAL GENERATION QUALITY

To study the impact of parallel multi-step denoising on generation quality, we evaluate the trained world models from Section 5.2 under varying decay horizon ($\nu$) and denoising budget ($B$) values. Our central question is how generation quality differs between parallel and sequential configurations, and which settings prove most effective.

In our evaluation, 512 episode segments of 33 frames are sampled from the training data. For each combination of $\nu$ and $B$, 512 corresponding 32-frame continuations are generated using the first frame as context. To isolate the impact of $\nu$ and $B$ from other factors such as the action sampling scheme, the original recorded actions are provided to the model at every denoising step. Generation quality is assessed using Fréchet Video Distance (FVD) (Unterthiner et al., 2019) and mean squared error (MSE) against the ground-truth segments. FVD better reflects perceptual quality, while MSE is more indicative of deviations from the ground truth sequence.

Our results (Figure 6) show a consistent FVD trend across environments, indicating a clear advantage for parallel configurations ($4 \leq \nu \leq 16$) under low to medium budgets. Remarkably, in several cases, parallel variants with sub-step budgets achieve quality comparable to baselines that rely on the maximal budget, 16 times larger. At extreme budgets ($B \geq 128$), performance tends to degrade slightly as $\nu$ increases, with the autoregressive baseline consistently ranking at the top. Interestingly,

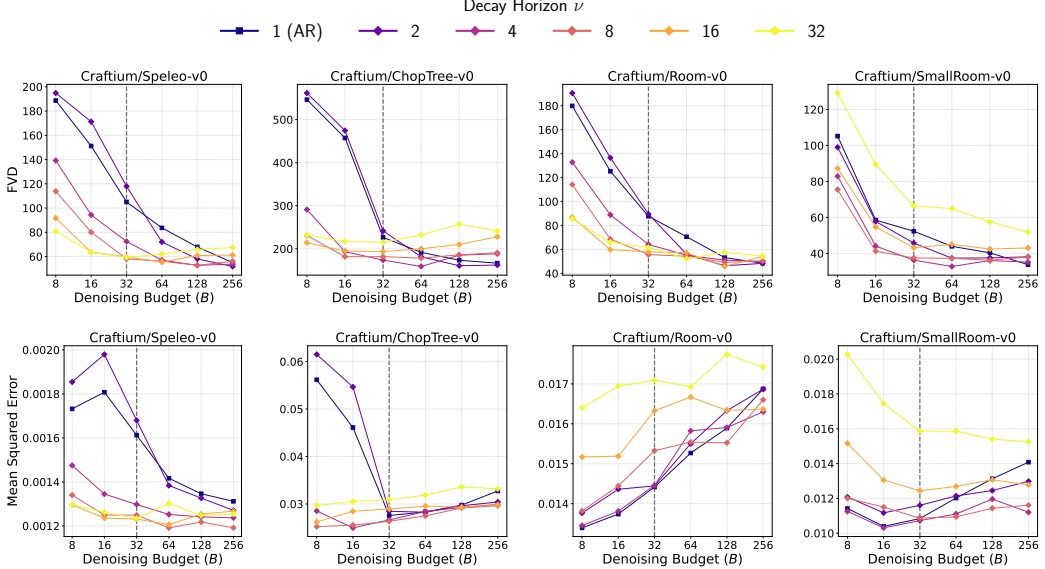

Figure 6: World model generation quality versus denoising steps budget. Each point shows the average FVD/MSE over 512 sampled 33-frame segments, where the first frame was given as context and the last 32 were generated conditioned on the recorded actions. A dashed vertical line indicates the transition out of sub-step budgets.

the MSE results (bottom row of Figure 6) suggest that increasing the denoising budget may cause generated sequences to drift further from the ground truth, even as perceptual quality improves.

Finally, in Appendix D, we present the corresponding Atari results and further repeat this experiment under the Pyramidal schedule of Chen et al. (2024), which suffers a pronounced collapse in performance as budgets grow.

## 6    LIMITATIONS

While our control results consistently highlight the advantage of $\nu = 4$, we did not extend our study to additional configurations. Evaluating each baseline requires running full experiments across all environments with five seeds per environment, which is computationally prohibitive. Consequently, we had to restrict our study to eight environments to keep the evaluations tractable. Future work could explore additional configurations.

## 7    CONCLUSIONS

In this work, we addressed the computational challenge of on-policy imagination in diffusion world models with discrete actions, motivated by the need to train lightweight policies for deployment. We introduced Horizon Imagination (HI), a parallel multi-step imagination procedure that incorporates two novel mechanisms: stable discrete action sampling and the Horizon schedule. Our analysis shows that stable sampling ensures consistent action selection, thereby stabilizing imagination, while the widely applicable Horizon schedule delivers sub-step performance and preserves robust decay horizons across budgets. Together, these advances allow HI to sustain strong performance even under sub-step budgets and at half the computational cost, making diffusion world models markedly more practical for training deployable policies.

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

## A  EXPERIMENTAL SETUP

### A.1  MODEL ARCHITECTURES

#### A.1.1  REPRESENTATION MODEL (TOKENIZER)

We use the continuous image autoencoder of Agarwal et al. (2025) as the representation model of our agent. We modify the output of the encoder network by adding a final $\tanh$ activation to enforce values in $[-1, 1]$. Empirically, we observed no loss of performance due to this modification. In addition, we adjust the configuration to fit our settings and computational budget. Further details on the architecture and hyperparameters can be found in Tables 1 and 3, and in our open-source code.

We use a standard mean squared error (MSE) regression objective for the reconstruction loss instead of the original $L_1$ objective, as we empirically found it to perform better in our settings. The full optimization objective is the sum of the reconstruction loss and a perceptual loss $\mathcal{L}_{\text{perceptual}}$ (Agarwal et al., 2025) (Eq. 3, p. 15). Formally, the loss is given by

$$\mathcal{L}_{\text{tokenizer}} = \|\hat{o} - o\|^2 + \mathcal{L}_{\text{perceptual}}, \tag{4}$$

where

$$\mathcal{L}_{\text{perceptual}} = \frac{1}{L} \sum_{l=1}^{L} \alpha_l \|\text{VGG}_l(\hat{o}) - \text{VGG}_l(o)\|_1,$$

is the perceptual loss, $\hat{o} = \varphi_{\text{dec}}(\varphi_{\text{enc}}(o))$ is the reconstructed observation produced by the autoencoder, $\text{VGG}_l$ are the features from the $l$-th layer of a pretrained VGG-19 network with $L$ layers, and $\alpha_l$ is a weight associated with the $l$-th loss term. We use the same hyperparameter values as Agarwal et al. (2025).

Table 1: The encoder and decoder architectures of the representation model. "Conv2d(a,b,c)" represents a 2d-convolutional layer with kernel size $a \times a$, stride of $b$ and padding $c$. "GN" represents a GroupNorm operator with 32 groups, $\epsilon = 1e - 6$ and learnable per-channel affine parameters. "Upsample" represents a repeat-interleave operation that doubles the spatial dimensions, followed by a Conv2d(3,1,1) layer.

| Module | Output Shape |
|---|---|
| Encoder | |
| Input | $3 \times 64 \times 64$ |
| Conv2d(3, 1, 1) | $64 \times 64 \times 64$ |
| ResnetBlock | $128 \times 64 \times 64$ |
| ResnetBlock | $128 \times 64 \times 64$ |
| Conv2d(3, 2, 0) | $128 \times 32 \times 32$ |
| ResnetBlock | $256 \times 32 \times 32$ |
| ResnetBlock | $256 \times 32 \times 32$ |
| Conv2d(3, 2, 0) | $256 \times 16 \times 16$ |
| ResnetBlock | $256 \times 16 \times 16$ |
| AttnBlock | $256 \times 16 \times 16$ |
| ResnetBlock | $256 \times 16 \times 16$ |
| AttnBlock | $256 \times 16 \times 16$ |
| Conv2d(3, 2, 0) | $256 \times 8 \times 8$ |
| ResnetBlock | $256 \times 8 \times 8$ |
| AttnBlock | $256 \times 8 \times 8$ |
| ResnetBlock | $256 \times 8 \times 8$ |
| GN | $256 \times 8 \times 8$ |
| Conv2d(3, 1, 1) | $16 \times 8 \times 8$ |
| Conv2d(1, 1, 0) | $16 \times 8 \times 8$ |
| $\tanh$ | $16 \times 8 \times 8$ |

| Module | Output Shape |
|---|---|
| Decoder | |
| Input | $16 \times 8 \times 8$ |
| Conv2d(1, 1, 0) | $16 \times 8 \times 8$ |
| Conv2d(3, 1, 1) | $256 \times 8 \times 8$ |
| ResnetBlock | $256 \times 8 \times 8$ |
| AttnBlock | $256 \times 8 \times 8$ |
| ResnetBlock | $256 \times 8 \times 8$ |
| ResnetBlock | $256 \times 8 \times 8$ |
| ResnetBlock | $256 \times 8 \times 8$ |
| ResnetBlock | $256 \times 8 \times 8$ |
| Upsample | $256 \times 16 \times 16$ |
| ResnetBlock | $256 \times 16 \times 16$ |
| AttnBlock | $256 \times 16 \times 16$ |
| ResnetBlock | $256 \times 16 \times 16$ |
| AttnBlock | $256 \times 16 \times 16$ |
| ResnetBlock | $256 \times 16 \times 16$ |
| AttnBlock | $256 \times 16 \times 16$ |
| Upsample | $256 \times 32 \times 32$ |
| ResnetBlock | $128 \times 32 \times 32$ |
| ResnetBlock | $128 \times 32 \times 32$ |
| ResnetBlock | $128 \times 32 \times 32$ |
| Upsample | $128 \times 64 \times 64$ |
| GN | $128 \times 64 \times 64$ |
| Conv2d(3, 1, 1) | $3 \times 64 \times 64$ |

### A.1.2 WORLD MODEL

Our world model consists of two components: the denoiser $v_\theta$ and a separate reward–termination predictor network. We view the dynamics model as a large-scale, general module that captures the complexity of the environment, whereas reward–termination predictors should remain lightweight, task-specific, and operate on the dynamics outputs. This design disentangles their dependencies: dynamics models can be scaled and pre-trained in advance, while downstream tasks require learning only small, efficient reward–termination predictors upon receiving a new task.

For the DiT-based denoiser $v_\theta$ (Peebles & Xie, 2023), we build on the Cosmos implementation (Agarwal et al., 2025) with several adaptations to our setting. As in Agarwal et al. (2025), inputs are patchified with a $2 \times 2$ spatial patch layer, and a corresponding reverse patchify layer is applied at the output. We also adopt 3D RoPE positional embeddings (Su et al., 2024; Agarwal et al., 2025), but in contrast to Agarwal et al. (2025), we omit any additional positional encodings.

Since no textual conditioning is required, we remove the cross-attention operator, yielding a vanilla DiT block structure. Attention is modified with a frame-level causal mask: tokens can attend to all other tokens in the same frame as well as to tokens from preceding frames. To condition on actions, we use a learned embedding table mapping discrete actions to vectors, which are summed with the denoising-time embeddings to form the conditioning input to the AdaLN layers.

Note that we only used the architecture implementation of Agarwal et al. (2025). Specifically, our work is based on the rectified flow diffusion framework (Liu et al., 2023), which is different from the EDM (Karras et al., 2022) framework used by Agarwal et al. (2025).

The reward–termination network processes latent observation sequences $\mathbf{z}_1, \ldots, \mathbf{z}_t$, analogous to $v_\theta$. Each latent is first encoded by a compact CNN consisting of a single ResNet block with 256 channels, followed by a $1 \times 1$ convolution that reduces the dimensionality to 64 channels. The resulting features are flattened and passed through a linear layer with a Sigmoid Linear Unit (SiLU) activation Hendrycks & Gimpel (2017); Ramachandran et al. (2018), yielding a vector representation for each 3D latent observation. Finally, the sequence of vectors is processed by an LSTM (Hochreiter & Schmidhuber, 1997) followed by two linear output heads that predict rewards and terminations, respectively.

The reward–termination model is trained on the same clean trajectory segments used for optimizing $v_\theta$. To accommodate reward signals that vary widely in scale and sparsity, we model rewards in symlog space. Formally, the reward objective is

$$\| r_\phi(\mathbf{z}_{\leq t+1}, \mathbf{a}_{\leq t}) - \mathrm{symlog}(r_t) \|^2,$$

where $r_\phi(\mathbf{z}_{\leq t+1}, \mathbf{a}_{\leq t})$ denotes the raw prediction and $\mathrm{symlog}(x) = \mathrm{sign}(x) \log(|x| + 1)$ is the symlog transform (Hafner et al., 2025). At inference time, the reward estimate is recovered via

$$\hat{r}_t = \mathrm{symexp}(r_\phi(\mathbf{z}_{\leq t+1}, \mathbf{a}_{\leq t})), \qquad \mathrm{symexp}(x) = \mathrm{sign}(x) \big( \exp(|x|) - 1 \big),$$

where $\mathrm{symexp}$ is the inverse of the symlog transform.

The termination predictor is trained using a standard cross-entropy loss.

### A.1.3 ACTOR-CRITIC

The critic $\hat{V}^\pi$ is trained exclusively on fully denoised outputs via the following regression objective in symlog space:

$$L_{\hat{V}^\pi} = \frac{1}{h} \sum_{t=1}^{h} \| \mathrm{symlog}(G_t) - \hat{V}^\pi(\mathbf{z}_{\leq t}, \mathbf{a}_{<t}) \|^2,$$

where $\mathrm{symlog}(x) = \mathrm{sign}(x) \log(|x| + 1)$ is the symlog function proposed by Hafner et al. (2025) for robust handling of values across scales and $G_t$ is the $\lambda$-return at step $t$:

$$G_t = \begin{cases} \bar{r}_t + \gamma(1 - d_t)((1 - \lambda) \mathrm{symexp}(\hat{V}_{t+1}^\pi) + \lambda G_{t+1}) & t < h, \\ \mathrm{symexp}(\hat{V}_h^\pi) & t = h. \end{cases}$$

Here, $\mathrm{symexp}(x) = \mathrm{sign}(x)(\exp(|x|) - 1)$ is the symlog inverse, $\hat{V}_t^\pi = \hat{V}^\pi(\mathbf{z}_{\leq t}, \mathbf{a}_{<t})$, and $\bar{r}_t$ is the reward generated by the learned reward model. We highlight that we found the above simple

regression objective in symlog space to perform robustly across rewards from dense to sparse and across reward scales, as can be seen in Figure 4.

Following Hafner et al. (2025), we scale the advantage terms using an exponential moving average (EMA):

$$A_t = \mathrm{sg}\left(\frac{G_t - \mathrm{symexp}(\hat{V}_t^\pi)}{\max(1, S)}\right),\tag{5}$$

where sg denotes the stop-gradient operator and

$$S = \mathrm{EMA}(\mathrm{quantile}(\boldsymbol{G}, 95) - \mathrm{quantile}(\boldsymbol{G}, 5), 0.005)$$

is the difference between the 95th and 5th return quantiles within the current batch, smoothed by an EMA with coefficient 0.005.

### A.2 HYPERPARAMETERS

For the actor-critic control performance evaluation experiments, we use the same agent hyperparameters across all environments and benchmarks. High-level agent training parameters are detailed in Table 2. The hyperparameters for the representation model (tokenizer), world model, and actor-critic, are presented in Tables 3, 4, and 5, respectively.

Table 2: Agent hyperparameters.

| Description | Value |
| --- | --- |
| Number of epochs | 500 |
| Data collection steps per epoch | 200 |
| Tokenizer optimization steps per epoch | 300 |
| Tokenizer train from epoch | 10 |
| World model optimization steps per epoch | 300 |
| World model train from epoch | 25 |
| Actor-critic optimization steps per epoch | 50 |
| Actor-critic train from epoch | 40 |

Table 3: Representation model (tokenizer) hyperparameters.

| Description | Value |
| --- | --- |
| Optimizer | AdamW |
| Learning rate | 2e-4 |
| AdamW weight decay | 0.05 |
| AdamW $(\beta_1, \beta_2)$ | (0.9, 0.95) |
| Batch size | 32 |
| Max. grad norm | 1 |
| Patch size | 1 |

Table 4: World model hyperparameters.

| Description | Value |
|---|---|
| Optimizer | AdamW |
| Learning rate | 2e-4 |
| AdamW weight decay | 0.01 |
| AdamW $(\beta_1, \beta_2)$ | (0.9, 0.99) |
| AdamW $\epsilon$ | 1e-6 |
| Max. grad norm | 1 |
| Batch size | 8 |
| Training horizon (generation length) | 32 |
| Denoiser $(v_\theta)$ Hyperparameters: | |
| Number of Transformer layers | 12 |
| Number of Attention heads | 8 |
| Attention head dimension | 64 |
| Embedding dimension | $512 = 64 \times 8$ |
| Spatial patch size | $2 \times 2$ |
| Reward-Termination Model Hyperparameters: | |
| CNN hidden channels | 256 |
| CNN output channels | 64 |
| LSTM hidden dimension | 512 |

Table 5: Actor-critic hyperparameters.

| Description | Value |
|---|---|
| Optimizer | AdamW |
| Learning rate | 2e-4 |
| AdamW weight decay | 0.01 |
| AdamW $(\beta_1, \beta_2)$ | (0.9, 0.99) |
| Max. grad norm | 1 |
| GAE $\lambda$ | 0.95 |
| GAE $\gamma$ (discount factor) | 0.99 |
| Entropy weight | 0.001 |
| Imagination actor-critic context length | 20 |
| Imagination world model context length | 1 |
| Imagination batch size | 30 |
| Imagination generation horizon | 32 |

For Atari, we follow the standard default setting with one exception: sticky actions are replaced by the max-initial no-ops wrapper for randomness, as in prior work (Micheli et al., 2023; Cohen et al., 2025; 2024; Alonso et al., 2024). Due to computational constraints, we were limited to a single evaluation across the benchmarks. Moreover, our agent's architecture was untested in a sample-efficiency setting, with no prior evidence to support its performance. We therefore adopted a well-established setting with a proven track record to mitigate risk. In hindsight, however, given the observed results, this precaution may not have been necessary. Nonetheless, unlike most prior works, we did not use sign rewards or termination on life loss. The Atari hyperparameters are presented in Table 6.

Table 6: Atari 100K hyperparameters.

| Description | Value |
|---|---|
| Frame resolution | $64 \times 64$ |
| Frame color space (RGB / grayscale) | RGB |
| Frame Skip | 4 |
| Max random initial no-ops | 30 |
| Sticky actions probability | 0.0 |
| Terminate on live loss | No |
| Sign rewards | No |

Table 7: Craftium hyperparameters.

| Description | Value |
|---|---|
| Frame resolution | $64 \times 64$ |
| Frame Skip | 4 |
| time speed | 0 (fixed time of day) |
| Sync mode | True |
| FPS max. (all excl. ChopTree-v0) | 200 |
| FPS max. (ChopTree-v0) | 10 |

## A.3 IMPLEMENTATION DETAILS

**Actor Initialization** Since exploration difficulty is not central to our evaluation, we initialize the actor network's biases with task-specific priors in select Craftium environments to encourage more effective data collection in the early stages, before actor–critic training begins. In `Craftium/Speleo-v0`, we set the forward action bias to 1 and all others to 0. In `Craftium/ChopTree-v0`, we set the dig (chop) action bias to 1 and the rest to 0.

**Replay Buffer Sampling** Under uniform sampling, as training progresses and the replay buffer grows, the probability of sampling recently collected data decreases. To mitigate this, we sample 70% of the batch uniformly, while the remaining 30% are sampled from a Beta$(3, 1)$ distribution (sample index is $i = \lfloor x \rfloor$, $x \sim$ Beta$(3, 1)$).

## B   RUNTIME ANALYSIS

Table 8 provides a detailed runtime analysis for the online training of the agent. Table 9 includes imagination throughput details.

Table 8: Breakdown of training stage durations (in milliseconds) for budgets $B = 16$ and $B = 32$ measured on an RTX 4090 GPU.

| Stage (Time in ms) | B = 16 | B = 32 |
|---|---|---|
| Tokenizer training step | 72.5 | 72.5 |
| World model training step | 133.2 | 133.2 |
| Controller training step | 1071 | 1871 |
| ∟ Imagination | 673 | 1326.5 |
| ∟ Reward prediction | 5.18 | 5.18 |
| ∟ Denoising total ($B\times$ steps) | 652.8 | 1305.6 |
| ∟ Single step duration | 40.8 | 40.8 |
| ∟ Action computation | 6.5 | 6.5 |
| ∟ Denoiser forward | 34.3 | 34.3 |
| **Epoch Total Time (sec)** | | |
| Tokenizer (300 steps) | 21.75 | 21.75 |
| World model (300 steps) | 39.96 | 39.96 |
| Controller imagination (50 steps) | 53.55 | 93.55 |

Table 9: Imagination throughput for budgets $B = 16$ and $B = 32$ under the configuration detailed in Appendix A, measured on an RTX 4090 GPU.

| | B = 16 | B = 32 |
|---|---|---|
| Frame denoising steps per second | 23529.4 | 23529.4 |
| Frame generations per second | 1470.6 | 735.3 |
| 32-frame segments per second | 46 | 23 |

## C  ALGORITHM PSEUDOCODE

---

**Algorithm 2** Agent Training Outline

---

1: **procedure** `training_loop`
2:     **repeat**
3:         `collect_experience(num_steps_to_collect)`
4:         **for** representation model update steps **do**
5:             `train_representation_model()`
6:         **end for**
7:         **for** world model update steps **do**
8:             `train_world_model()`
9:         **end for**
10:        **for** controller update steps **do**
11:           `train_controller()` (Alg. )
12:        **end for**
13:     **until** stopping criterion is met (e.g., target number of epochs)
14: **end procedure**

---

15: **procedure** `collect_experience(`$n$`)`
16:     Initialize controller state with latest context (real env.)
17:     **for** $t = 0$ to $n$ **do**
18:         Sample action $a_t \sim \pi(a_t | \mathbf{z}_{\leq t})$
19:         Apply the action in the environment: $\mathbf{o}_{t+1}, r_t, d_t \leftarrow$ `env.step(`$a_t$`)`
20:         Store experience in replay buffer
21:         **if** $d == 1$ **then**
22:             Reset environment and model context.
23:         **end if**
24:     **end for**
25: **end procedure**

---

26: **procedure** `train_representation_model`
27:     Sample a batch of (independent) observations (frames) $\mathbf{o}$ from the replay buffer
28:     Compute encoder outputs $\mathbf{z} = \varphi_{\text{enc}}(\mathbf{o})$
29:     Compute decoder outputs (reconstructions) $\hat{o} = \varphi_{\text{dec}}(\mathbf{z})$
30:     Compute loss (Eq. 4, Section A.1.1)
31:     Update model weights
32: **end procedure**

---

33: **procedure** `train_world_model`
34:     Sample a batch of trajectory segments from the replay buffer
35:     Compute latent observation representations $\mathbf{z}_t = \varphi_{\text{enc}}(\mathbf{o}_t)$ for all observations
36:     Sample corresponding noise sequences $\mathbf{z}^0$ and denoising times $\tau$
37:     Compute noisy sequences $\mathbf{z}^\tau = \tau \mathbf{z}^1 + (1 - \tau) \mathbf{z}^0$
38:     Compute denoiser outputs $v_\theta(\mathbf{z}^\tau, \tau, a)$ (single forward pass)
39:     Compute denoiser loss (Eq. 1)
40:     Compute reward-termination loss (Section A.1.2)
41:     Update models weights
42: **end procedure**

---

43: **procedure** `train_controller`
44:     Sample a batch of trajectory segments from the replay buffer (context)
45:     Initialize controller state and set world model context
46:     Generate future trajectory using Horizon Imagination (Alg. 1)
47:     Compute control objectives (Sections 4.3 and A.1.3)
48:     Update controller weights
49: **end procedure**

---

# D  ADDITIONAL RESULTS

**Parallel vs. Sequential Generation Quality in Atari**  Figure 7 presents additional generation quality evaluations for the Atari benchmark. Notably, we observe the same overall trends while FVD values are generally lower across configurations due to the visually simpler observation space and dynamics. The MSE values are orders of magnitude smaller, which aligns with the fact that the majority of pixels remain fixed between frames in Atari.

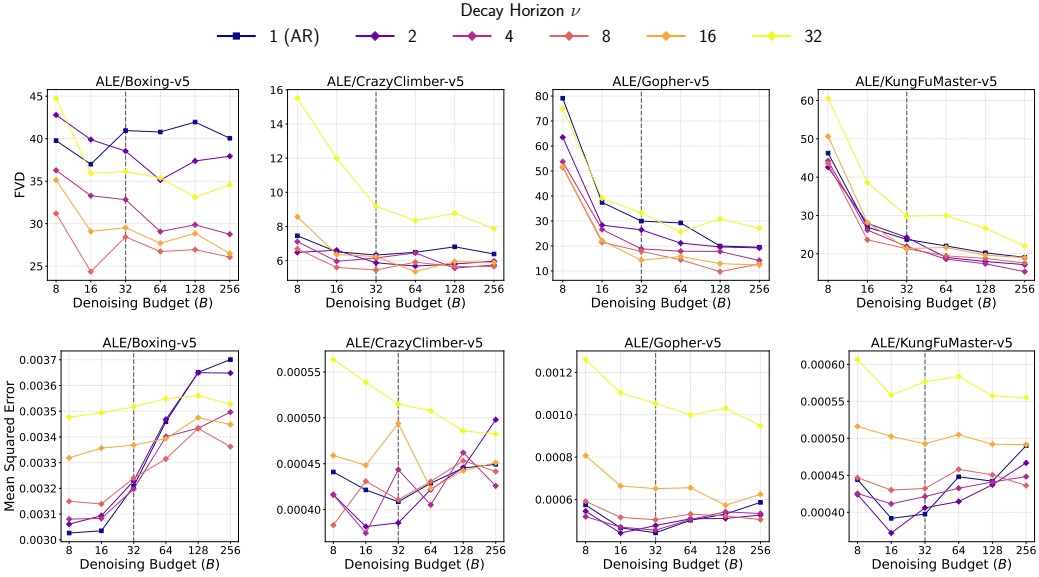

Figure 7: World model generation quality versus denoising steps budget for the subset of Atari games. Each point shows the average FVD/MSE over 512 sampled 33-frame segments, where the first frame was given as context and the last 32 were generated conditioned on the recorded actions. A dashed vertical line indicates the transition out of sub-step budgets.

**Pyramidal Schedule (Diffusion Forcing) Performance Degradation**  Figure 8 reports results under the same experimental setup as Section 5.3, except using the Pyramidal schedule of Chen et al. (2024). Our findings reveal significant degradation at high budgets, driven by the coupling of budget and decay horizon. Instead of improving with more computation, generation quality collapses as the budget grows. Furthermore, unlike our schedule, this approach lacks support for sub-step budgets.

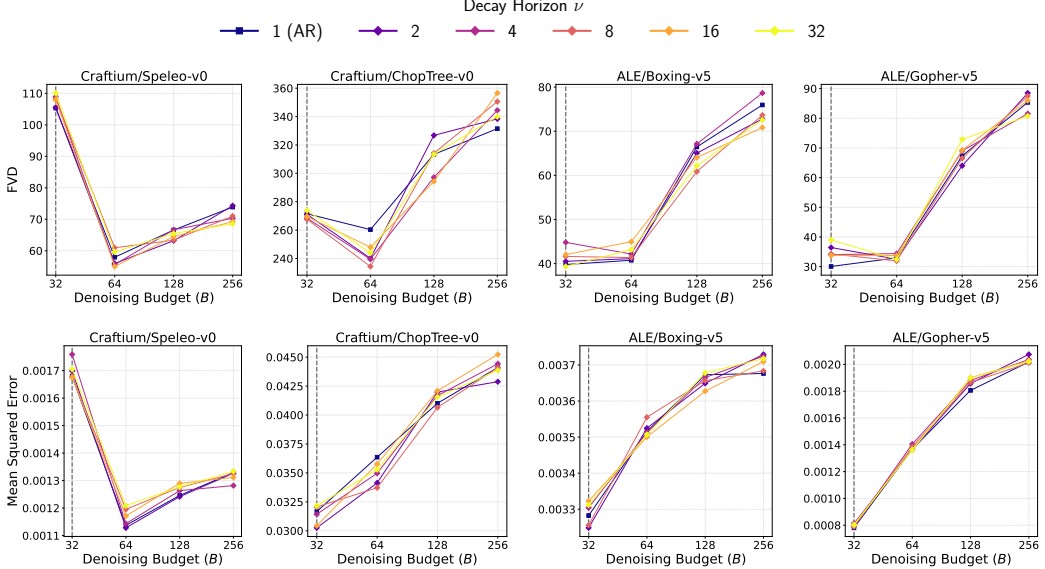

Figure 8: World model generation quality versus denoising steps budget when using the Pyramidal schedule of Chen et al. (2024). Each point shows the average FVD/MSE over 512 sampled 33-frame segments, where the first frame was given as context and the last 32 were generated conditioned on the recorded actions. A dashed vertical line indicates the transition out of sub-step budgets.

**Qualitative Visualizations of Horizon Schedule Configurations**   To complement the quantitative results in Section 5.3, we present qualitative visualizations of the generated sequences used in that analysis, across different configurations of the proposed Horizon schedule. From the large set of possible configurations and samples, we select a subset that highlights the most important cases, reflects overall trends, and also provides a visual impression of the environment and the task. We further show ground-truth frames (GT) and their tokenizer reconstructions (Rec) to evaluate representation quality in isolation from the world model. This enables us to disentangle errors due to imperfect representations from those caused by dynamics modeling. Representative examples are provided in Figures 9-16.

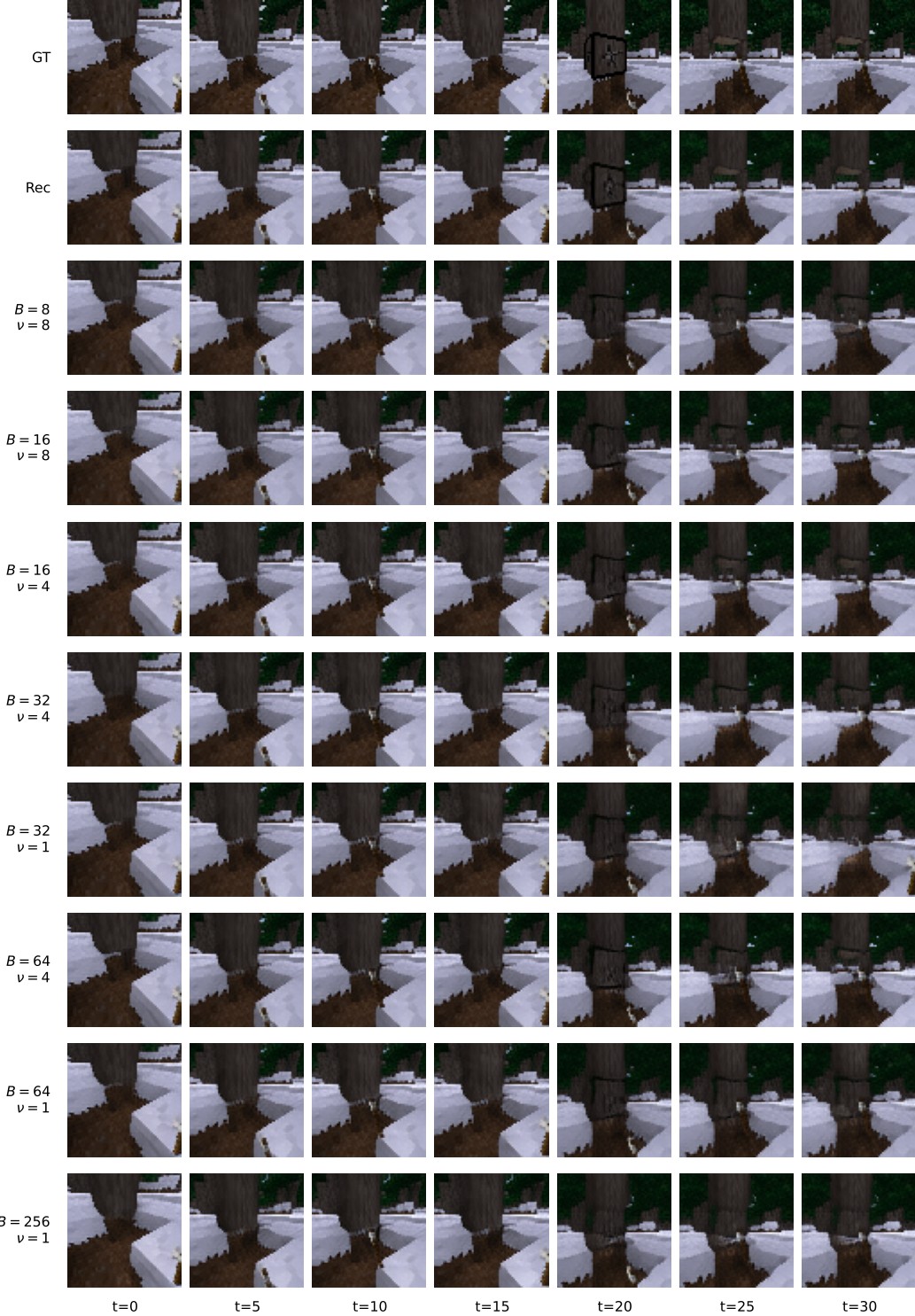

Figure 9: Qualitative visualization of generated sequences in `Craftium/ChopTree-v0` under different Horizon schedule configurations $(B, \nu)$. Ground-truth frames (GT) and their tokenizer reconstructions (Rec) are provided for reference. Configurations with $\nu = 1$ correspond to the autoregressive baseline.

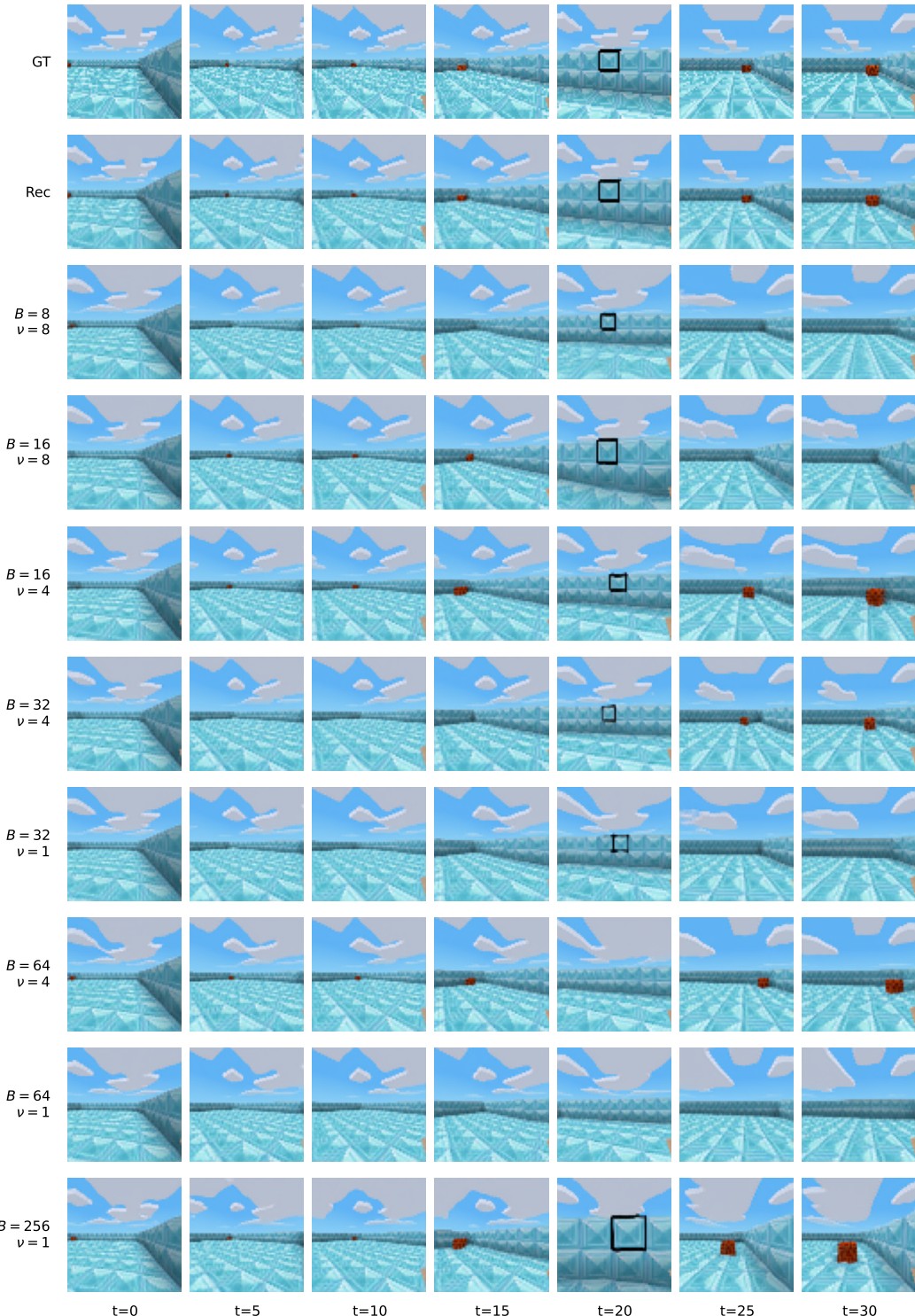

Figure 10: Qualitative visualization of generated sequences in `Craftium/Room-v0` under different Horizon schedule configurations $(B, \nu)$. Ground-truth frames (GT) and their tokenizer reconstructions (Rec) are provided for reference. Configurations with $\nu = 1$ correspond to the autoregressive baseline.

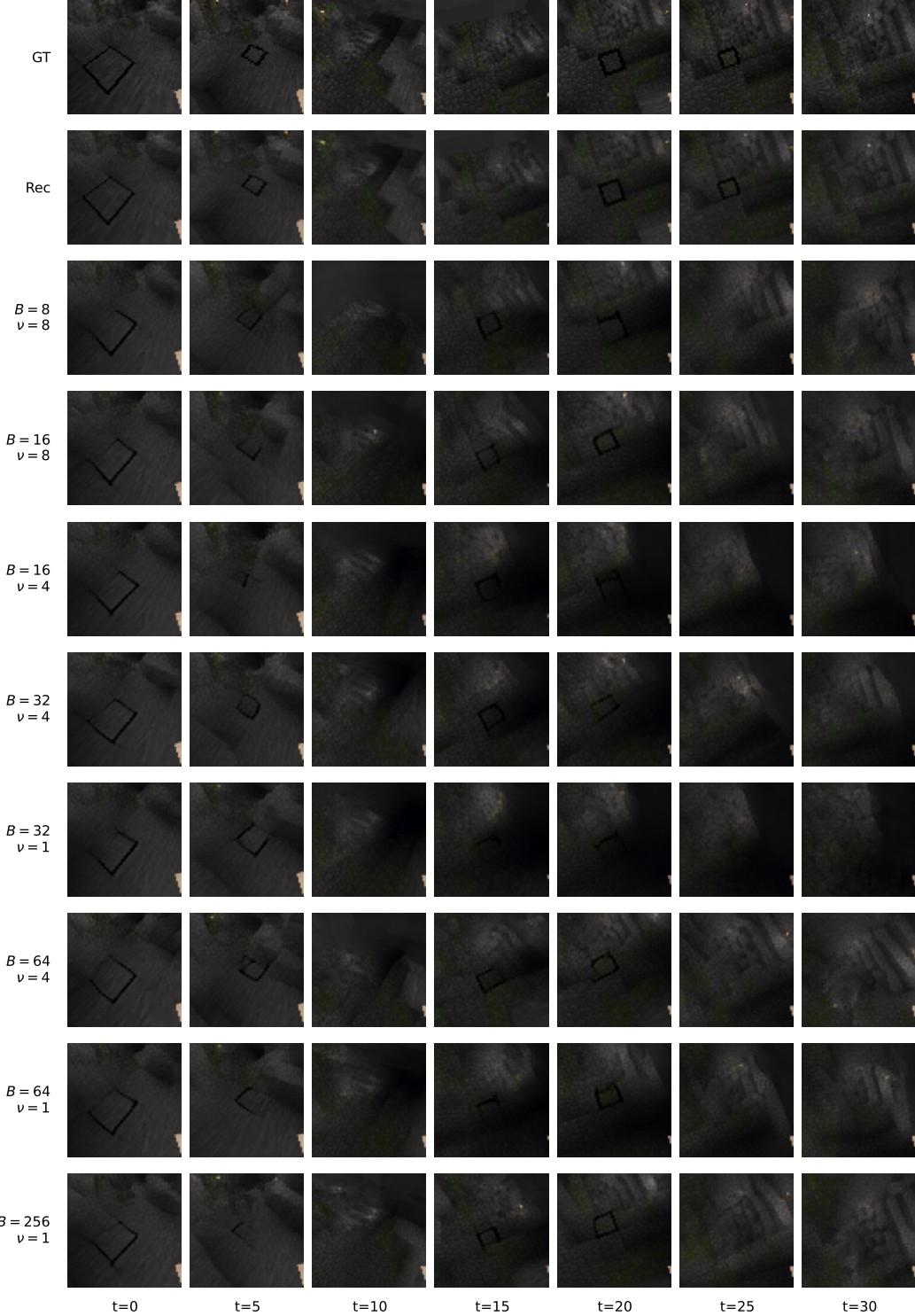

Figure 11: Qualitative visualization of generated sequences in `Craftium/Speleo-v0` under different Horizon schedule configurations $(B, \nu)$. Ground-truth frames (GT) and their tokenizer reconstructions (Rec) are provided for reference. Configurations with $\nu = 1$ correspond to the autoregressive baseline.

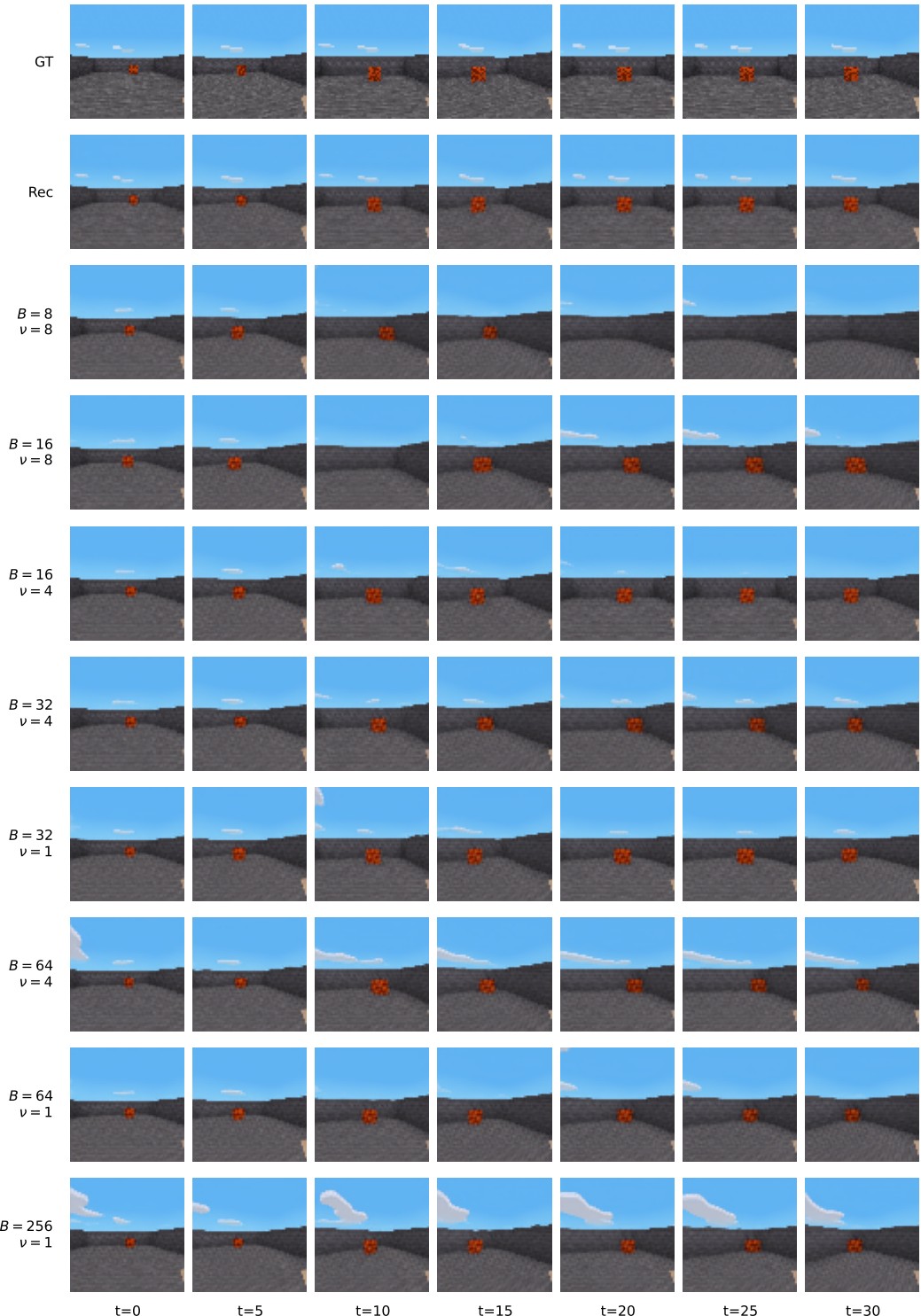

Figure 12: Qualitative visualization of generated sequences in `Craftium/SmallRoom-v0` under different Horizon schedule configurations $(B, \nu)$. Ground-truth frames (GT) and their tokenizer reconstructions (Rec) are provided for reference. Configurations with $\nu = 1$ correspond to the autoregressive baseline.

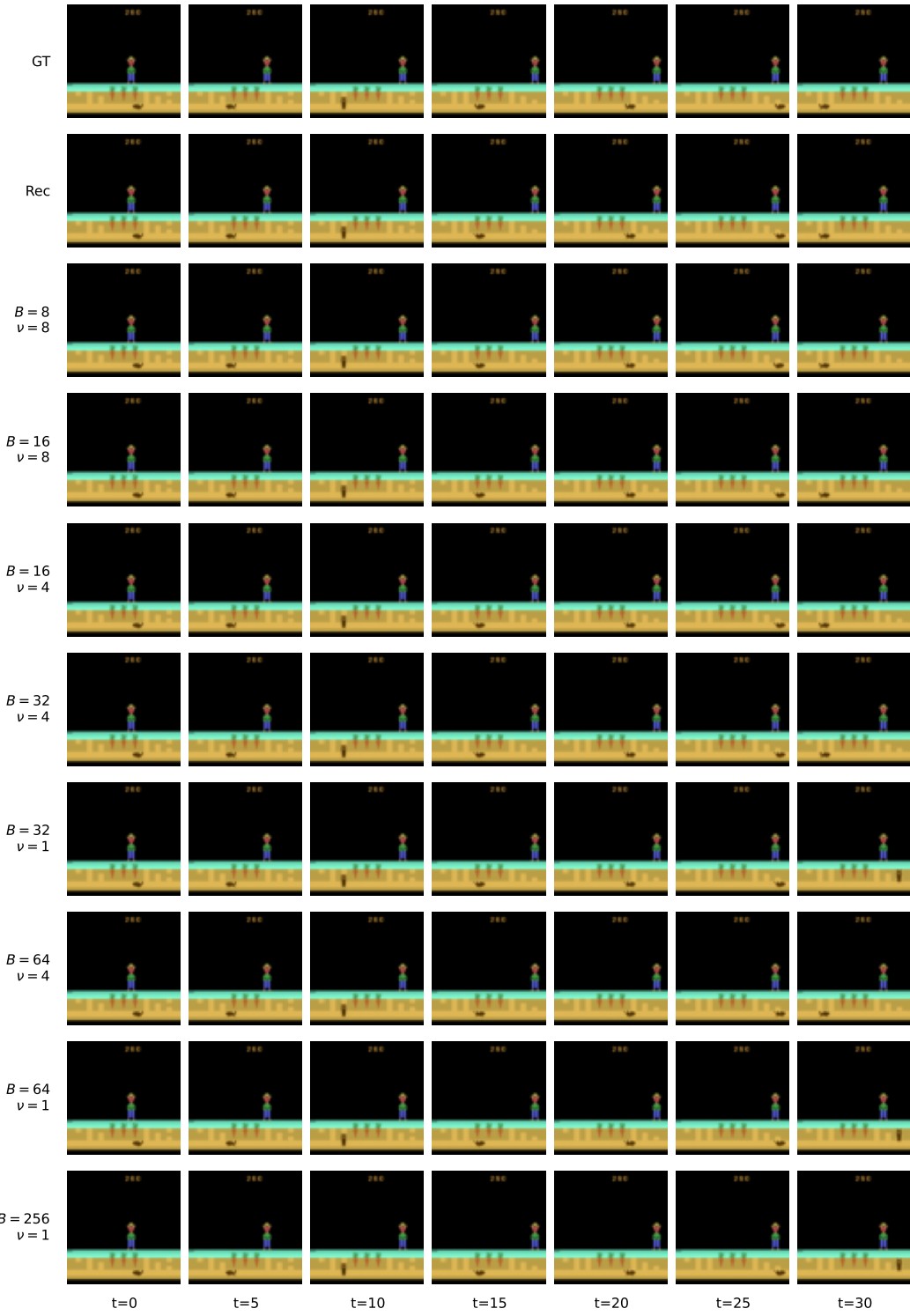

Figure 13: Qualitative visualization of generated sequences in ALE/Gopher-v5 under different Horizon schedule configurations $(B, \nu)$. Ground-truth frames (GT) and their tokenizer reconstructions (Rec) are provided for reference. Configurations with $\nu = 1$ correspond to the autoregressive baseline.

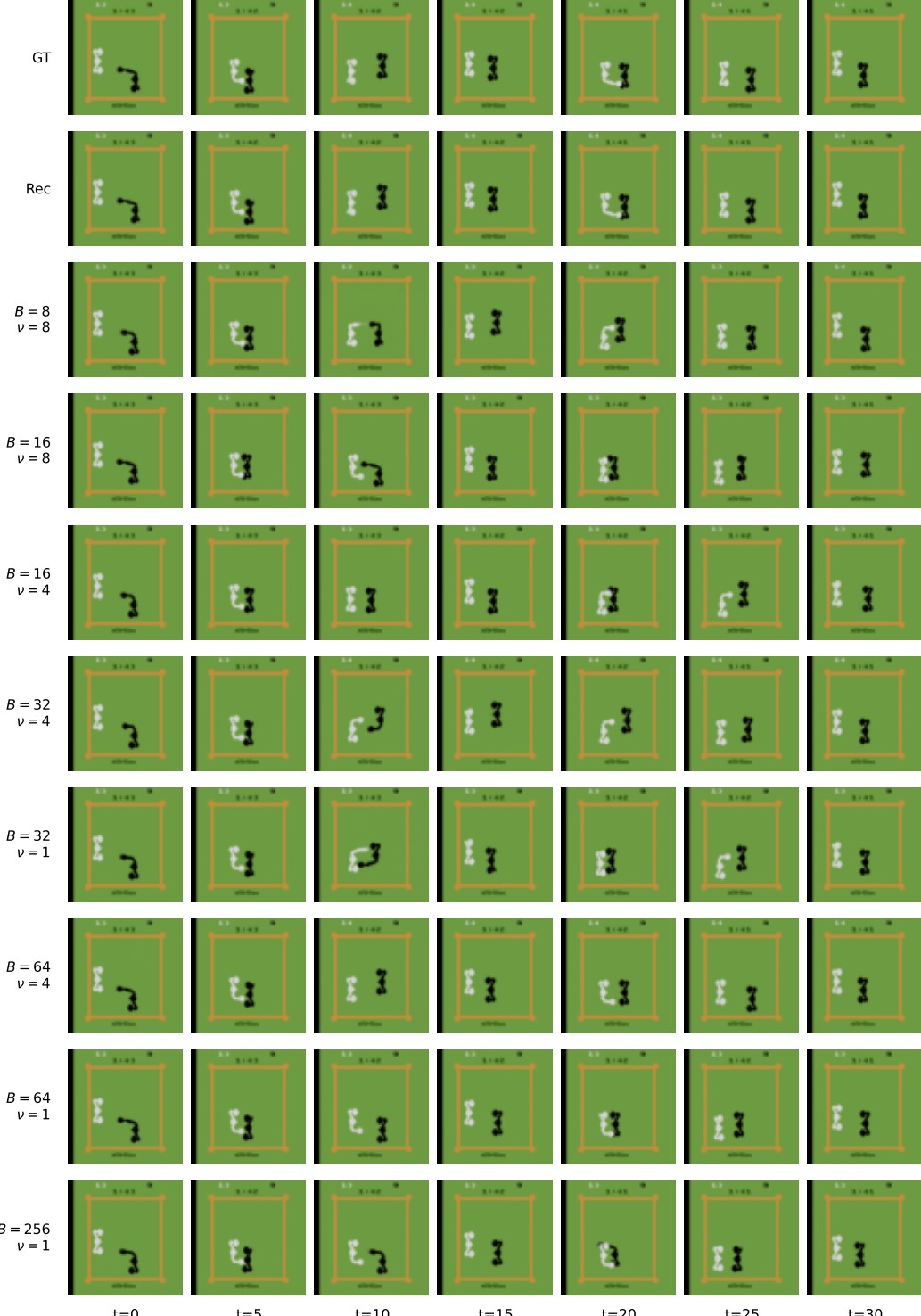

Figure 14: Qualitative visualization of generated sequences in ALE/Boxing-v5 under different Horizon schedule configurations $(B, \nu)$. Ground-truth frames (GT) and their tokenizer reconstructions (Rec) are provided for reference. Configurations with $\nu = 1$ correspond to the autoregressive baseline.

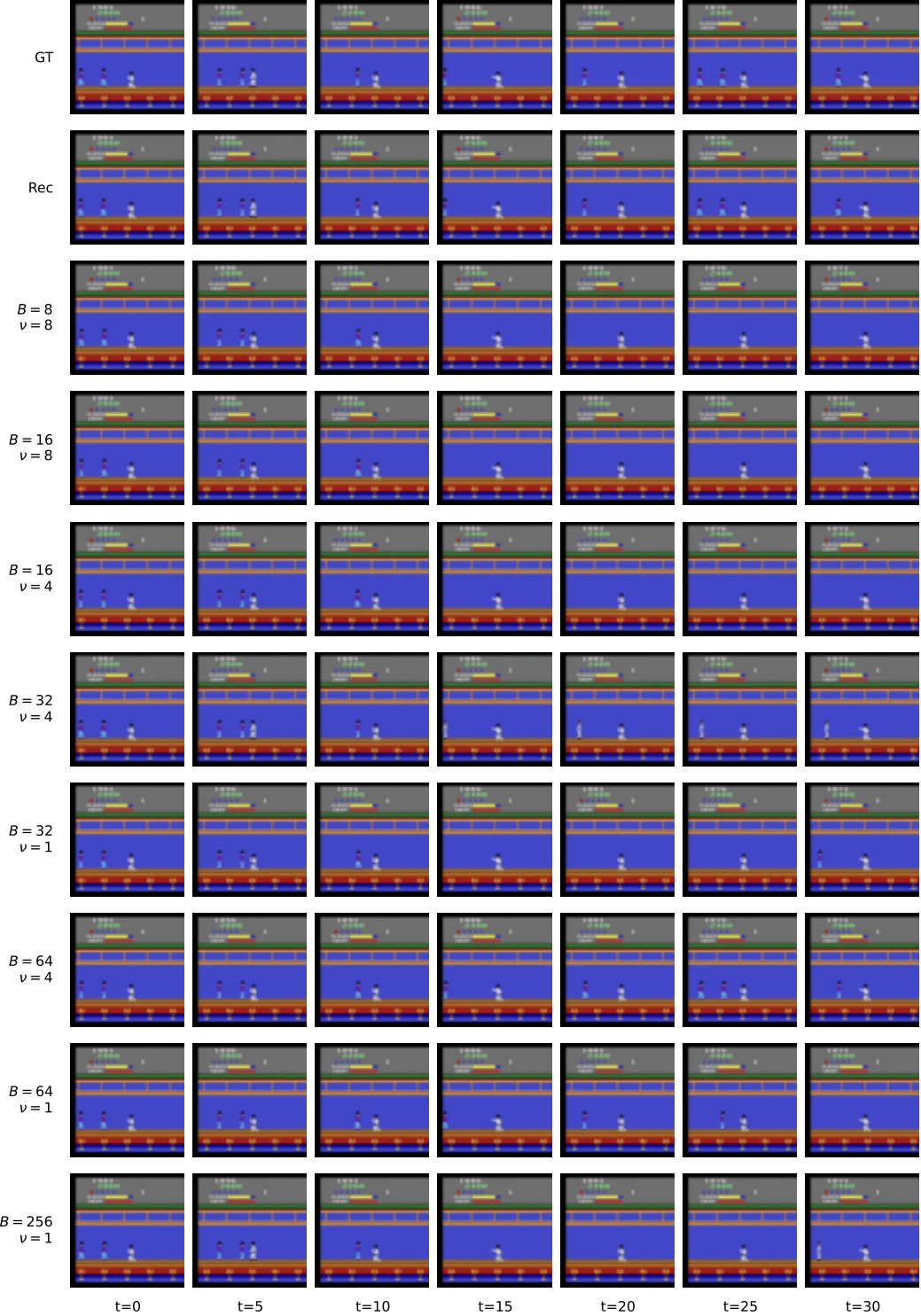

Figure 15: Qualitative visualization of generated sequences in `ALE/KungFuMaster-v5` under different Horizon schedule configurations $(B, \nu)$. Ground-truth frames (GT) and their tokenizer reconstructions (Rec) are provided for reference. Configurations with $\nu = 1$ correspond to the autoregressive baseline.

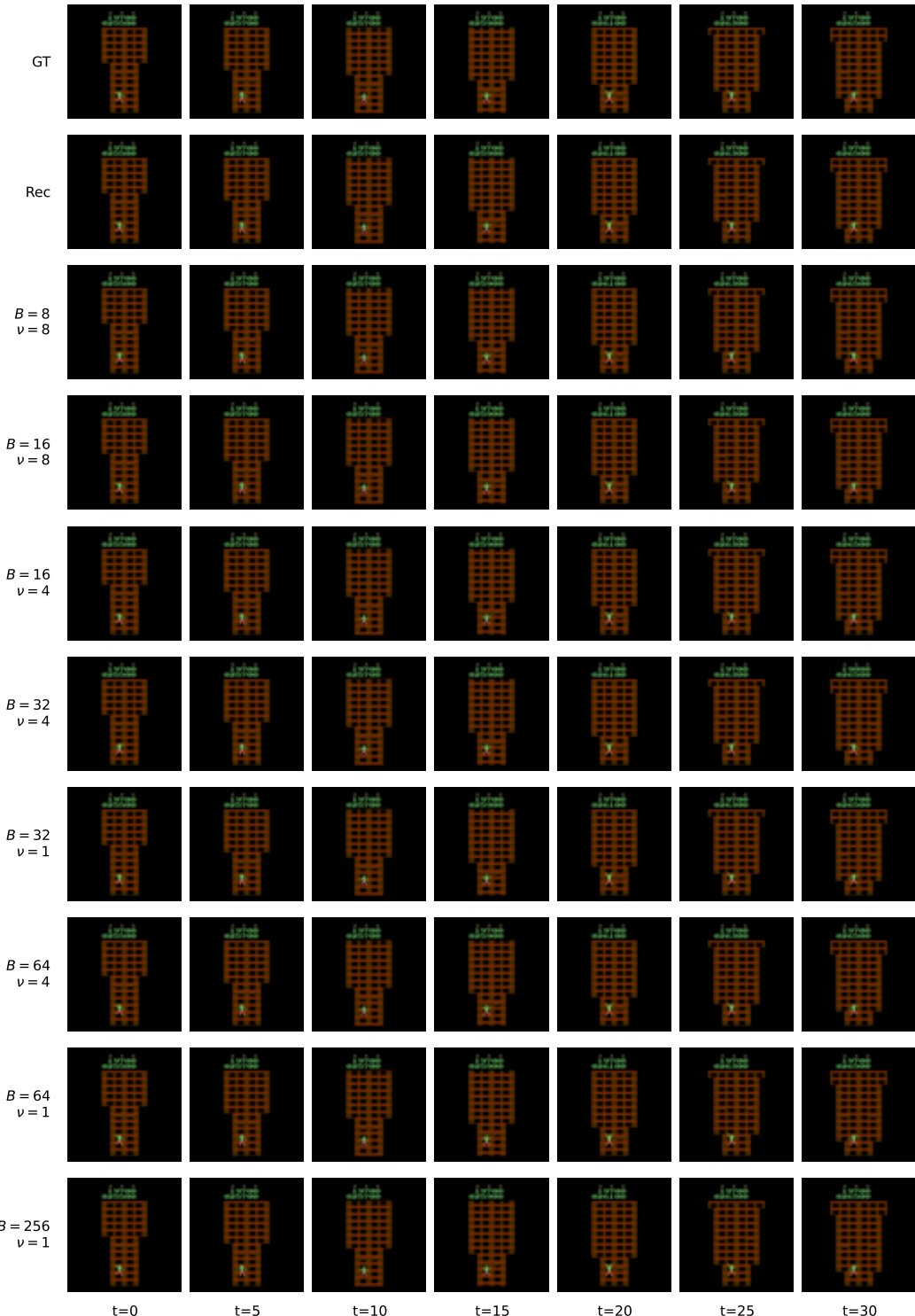

Figure 16: Qualitative visualization of generated sequences in `ALE/CrazyClimber-v5` under different Horizon schedule configurations $(B, \nu)$. Ground-truth frames (GT) and their tokenizer reconstructions (Rec) are provided for reference. Configurations with $\nu = 1$ correspond to the autoregressive baseline.

# E    PROOF FOR PROPOSITION 1

## E.1    PROOF TO PROPOSITION 1.1

*Proof.* Let $\boldsymbol{p} \in \Delta^{N-1}$. Without loss of generality, let $\rho = (1, 2, \ldots, N)$ so that $\rho(i) = i$, which simplifies notations.

Let $1 \leq i < N$. If $S_i(\boldsymbol{p}) := \sum_{j=i}^{N} p_j = 0$, then by the definition of $\alpha_i$ it holds that

$$\Pr(\alpha(\boldsymbol{p}, \cdot) = i) = \Pr(\omega_i < \alpha_i(\boldsymbol{p})) = \Pr(\omega_i < p_i) = p_i.$$

Otherwise, $S_i(\boldsymbol{p}) > 0$. In that case,

$$\Pr(\mathrm{a}(\boldsymbol{p}, \cdot) = i) = \Pr(\omega_i < \alpha_i(\boldsymbol{p})) \prod_{j=1}^{i-1} \Pr(\omega_j \geq \alpha_j(\boldsymbol{p}))$$

$$= \frac{p_i}{\sum_{k=i}^{N} p_k} \prod_{j=1}^{i-1} (1 - \alpha_j(\boldsymbol{p}))$$

$$= \frac{p_i}{\sum_{k=i}^{N} p_k} \prod_{j=1}^{i-1} \left(1 - \frac{p_j}{\sum_{k=j}^{N} p_k}\right)$$

$$= \frac{p_i}{\sum_{k=i}^{N} p_k} \prod_{j=1}^{i-1} \left(\frac{\sum_{k=j+1}^{N} p_k}{\sum_{k=j}^{N} p_k}\right)$$

$$= \frac{p_i}{\sum_{k=i}^{N} p_k} \left(\frac{\sum_{k=2}^{N} p_k}{\sum_{k=1}^{N} p_k}\right) \left(\frac{\sum_{k=3}^{N} p_k}{\sum_{k=2}^{N} p_k}\right) \cdots \left(\frac{\sum_{k=i}^{N} p_k}{\sum_{k=i-1}^{N} p_k}\right)$$

$$= p_i.$$

In the last equality, we used the fact that $\sum_{k=1}^{N} p_k = 1$. The probability of the last action follows from the fact that the probability mass sums to 1. $\square$

## E.2    PROOF TO PROPOSITION 1.2

*Proof.* The first inequality

$$\frac{1}{2} \|\boldsymbol{p} - \boldsymbol{q}\|_1 \leq \Pr(A) = \Pr(\mathrm{a}(\boldsymbol{p}, \omega) \neq \mathrm{a}(\boldsymbol{q}, \omega)) \quad \forall \omega \in [0, 1)^{N-1}$$

follows from the coupling lemma (Aldous, 1983) (Lemma 3.6, p.249) and from Proposition 1.1.

For the second inequality $\Pr(A) \leq \|\alpha(\boldsymbol{p}) - \alpha(\boldsymbol{q})\|_1$, let

$$\boldsymbol{u} = \max(\alpha(\boldsymbol{p}), \alpha(\boldsymbol{q})) = (\max(\alpha_1(\boldsymbol{p}), \alpha_1(\boldsymbol{q})), \ldots, \max(\alpha_{N-1}(\boldsymbol{p}), \alpha_{N-1}(\boldsymbol{q}))),$$
$$\boldsymbol{l} = \min(\alpha(\boldsymbol{p}), \alpha(\boldsymbol{q})) = (\min(\alpha_1(\boldsymbol{p}), \alpha_1(\boldsymbol{q})), \ldots, \min(\alpha_{N-1}(\boldsymbol{p}), \alpha_{N-1}(\boldsymbol{q}))).$$

Without loss of generality, let $\rho = (1, 2, \ldots, N)$ so that $\rho(i) = i$, which simplifies notations. Recall that

$$A = \{\mathrm{a}(p, \omega) \neq \mathrm{a}(q, \omega)\}, \quad \omega \sim \mathcal{U}([0, 1))^{N-1}.$$

Consider the following partition of $[0, 1)^{N-1}$:

$$[0, 1)^{N-1} = \left[\bigcup_{i=1}^{N-1} (L_i \cup C_i)\right] \cup U,$$

$$L_i = \{\omega \mid \forall j < i : \omega_j \geq u_j \text{ and } \omega_i < l_i\},$$
$$C_i = \{\omega \mid \forall j < i : \omega_j \geq u_j \text{ and } \omega_i \in [l_i, u_i)\},$$
$$U = \{\omega \mid \forall 1 \leq i \leq N - 1 : \omega_i \geq u_i\}.$$

Notice that by the definition of $a(\cdot, \cdot)$, it holds that $a(\boldsymbol{p}, \omega) = a(\boldsymbol{q}, \omega)$ for all $\omega \in \left[ \bigcup_{i=1}^{N-1} L_i \right] \cup U$. Similarly, $\Pr(A) = \Pr(\omega \in \bigcup_{i=1}^{N-1} C_i)$. Hence,

$$
\begin{aligned}
\Pr(A) &= \sum_{i=1}^{N-1} \Pr(C_i) \\
&= \sum_{i=1}^{N-1} (u_i - l_i) \Pi_{j=1}^{i-1}(1 - u_j) \\
&= \sum_{i=1}^{N-1} |\alpha_i(\boldsymbol{p}) - \alpha_i(\boldsymbol{q})| \, \Pi_{j=1}^{i-1}(1 - u_j) \\
&\leq \sum_{i=1}^{N-1} |\alpha_i(\boldsymbol{p}) - \alpha_i(\boldsymbol{q})| \\
&= \|\alpha(\boldsymbol{p}) - \alpha(\boldsymbol{q})\|_1
\end{aligned}
$$

$\square$

## F  POTENTIAL EXTENSION TO CONTINUOUS ACTION SPACES

Although the stable action sampling mechanism in Horizon Imagination is tailored to discrete action spaces, we propose a direct extension to continuous domains that we leave for future investigation. Specifically, consider a Gaussian policy with continuous outputs

$$\mu_t, \sigma_t = \pi(\cdot | \mathbf{z}_{\leq t}, \mathbf{a}_{<t}).$$

To adapt the method to the continuous case, we consider two modifications. First, the initial sample distribution is drawn from a standard normal, $\omega \sim \mathcal{N}(\mathbf{0}, \mathbf{I})$. Second, using the reparameterization trick, we define the action mapping as

$$\mathbf{a}(\pi, \omega) = \mu + \omega \sigma.$$

This construction would minimize action changes when the policy distribution parameters $\mu, \sigma$ remain fixed, reduce action differences when they vary, and ensure that $\mathbf{a}(\pi, \omega)$ follows the policy's distribution, as in the discrete case. Notably, such an extension may also avoid the computational overhead associated with classifier-guidance methods used in prior work.

## G  DECOUPLED CLEAN AND NOISY ACTOR VARIANT

During additional development experiments on Atari Breakout, we observed that the actor update used in the main paper underperforms on this game. We did not observe similar degradation in the environments evaluated in the main experiments. Motivated by this observation, we explored a decoupled actor variant that maintains separate clean and noisy policy copies.

The clean actor is used for interaction with the real environment during data collection, and is updated during the imagination phase using the same actor–critic objective as in the main method (REINFORCE with advantage estimation and entropy regularization), computed on fully denoised imagined rollouts.

The noisy actor is used exclusively during world-model imagination, where it is invoked at each denoising step and actions are selected using our stable sampling mechanism. After rollout generation, the noisy actor is trained via cross-entropy so that, given noisy latent observations, it matches the action distribution produced by the clean actor on the corresponding clean latents.

Preliminary experiments indicate improved performance on Breakout and promising behavior on a small number of additional Atari games. Due to compute constraints, we did not re-run the full benchmark suite with this variant, and leave a comprehensive evaluation to future work.

## H  USE OF LARGE LANGUAGE MODELS (LLMS)

In our work, we used LLMs for the following tasks:

1. Aid and polish writing. This includes revising original drafts of sentences or short paragraphs for improved phrasing.
2. Writing python scripts that produce matplotlib plots from raw results. This includes manipulating existing results in raw data formats (e.g., ".csv", ".json", images, videos), and quickly drawing high quality plots using matplotlib.

