# OpenReview forum: "Horizon Imagination: Efficient On-Policy Rollout in Diffusion World Models"
_ICLR.cc/2026/Conference — ICLR 2026 Poster_

### Official Review · Reviewer_8WiC · 2025-10-26

**Soundness:** 3
**Presentation:** 2
**Contribution:** 3
**Rating:** 6
**Confidence:** 2

**Summary:**

To address the low sampling efficiency of existing diffusion world models, especially the quadratic computational cost arising from the product of environment steps and diffusion steps, the authors introduce Horizon Imagination, an algorithm that enables more efficient on-policy sampling of discrete stochastic policies within diffusion world models. The authors make two key improvements over prior diffusion model sampling methods. First, they introduce a horizon sampling schedule that separates the total denoising budget from the decay horizon, allowing flexible and effective configuration. They also propose a novel sampling strategy that stabilizes policy output actions during denoising and prove it preserves the original sampling distribution. Experiments on Atari 100K and Craftium show that Horizon Imagination greatly reduces denoising steps and outperforms previous schedules like pyramidal schedule in generation quality.

**Strengths:**

1. The diffusion-based world model studied in this paper is a rapidly growing research area with significant value in both offline data generation and online policy learning. It holds great promise for substantially reducing the cost of real-world interactions.

2. The authors effectively resolve the instability that occurs when diffusion models and policies interact to jointly sample multi-step trajectories by introducing a theoretically grounded stable action sampling method. This approach significantly enhances the quality and stability of long-horizon trajectory generation.

3. Compared with the previous pyramidal schedule, the proposed horizon schedule provides a more general formulation. By decoupling the decay horizon from the denoising budget, it enables more consistent denoising schedules and achieves higher-quality generation.

**Weaknesses:**

1. The stable action sampling algorithm proposed by the authors is only applicable to discrete action spaces, which limits the applicability of the Horizon Imagination framework in more general environments with continuous action spaces.

2. A key weakness is the limited scope of the experimental comparisons. The control performance results in Section 5.2 are structured as an internal ablation study, comparing the proposed parallel method only against an autoregressive baseline within their own framework. The paper does not benchmark its end-to-end performance against other established world model agents, such as those discussed in the related work. The absence of direct, end-to-end SOTA comparisons makes it difficult to fully assess the proposed method's relative performance and efficiency in the field.

3. Training for 19 or 27 hours on A100 for relatively simple games like Atari seems somewhat costly. Conducting experiments in more challenging environments could make the proposed method more practically valuable.

4. There is an extra quotation mark at the end of line 446.

**Questions:**

1. Are there any possible exploratory directions for extending the idea of Horizon Imagination to continuous action spaces?

2. Algorithm 1 only includes the world model sampling process. Could the authors provide a complete pseudocode of the entire training pipeline, including world model training and RL algorithm updates? This would help readers gain a clearer understanding of the overall framework.

---

> ### Author Response · Authors · 2025-11-19
>
> We thank reviewer 8WiC for the time invested in evaluating our work, for their thoughtful comments, and for the positive overall assessment.
>
>
> We are currently revising the paper to incorporate the changes discussed below. The updated version will be available before the end of the discussion phase, and we will issue a notification when it is posted.
>
>
> #### Weaknesses
> `W1`:
> > The stable action sampling algorithm ...
>
> We focus on the discrete case because (i) multiple solutions already exist for continuous actions but none for discrete actions, (ii) typical applications use either discrete or continuous actions but not both simultaneously.
>
> As we discuss below (`A1`), there is a straightforward extension of our stable action sampling method to the continuous case, which future works can explore. We propose to include it in a future directions discussion.
>
>
> ---
>
>
> `W2`:
> > A key weakness is the limited scope...
>
> The control experiments in Section 5.2 are design to support the claim that Horizon imagination enables improved training/generation efficiency, compared to the standard auto-regressive approach it aims to replace.
>
> Importantly, absolute control performance are irrelevant for that purpose (assuming reasonable performance), as improving efficiency is an orthogonal objective. Therefore, including external baselines would not contribute toward supporting our argument. Note that we never claimed to improve the SOTA, or to improve absolute control performance, as this is not the focus of our paper.
>
> The appropriate experimental design for supporting our claim is thus to compare a standard control method (in our case, the actor-critic method we adopted from prominent prior works [1][2][3][4]) when trained with standard autoregressive imagination versus the proposed Horizon imagination.
>
> In addition, to the best of our knowledge, our work is the first to address parallel world model on-policy generation in the discrete action space setting. Hence, there are no relevant baselines to compare to.
>
>
> ---
>
>
> `W3`:
> > Training for 19 or 27 hours on A100 for ...
>
>
> **Training cost in the context of diffusion world models.**
> Diffusion-based world models in the sample-efficient RL regime (e.g., Atari 100K) are still very recent. Prior work in this space (e.g., [2]) relies on architectures with short context windows, limited scaling potential, and high computational requirements. In contrast, we adapt a Diffusion Transformer (DiT), which naturally supports long contexts, scales more efficiently, and in our experience is significantly cheaper to train and to sample from.
>
> Concretely, the architecture in [2] requires **~2.5 days on an RTX 4090**, whereas our agent trains in **~14.5 hours on the same GPU** (with $𝐵=16$). Thus, although diffusion models remain relatively costly, our method substantially reduces compute requirements compared to existing diffusion world-model agents. To the best of our knowledge, it is currently the fastest diffusion-based world-model agent in the sample-efficient RL setting.
>
> We open-source all code to facilitate further efficiency improvements by the community.
>
> **Regarding the more challenging environments concern:**
> Our work includes two main contributions:
> (1) improving efficiency (maintaining performance at a lower computational cost / attaining a better performance-cost relationship), and
> (2) identifying and correcting a design limitation in the Pyramidal schedule.
>
> The difficulty of the benchmark does not directly affect the validity of these claims, since an efficiency improvement manifests as reduced training cost regardless of whether the environment is easy or hard.
>
> Nevertheless, we chose to evaluate on Atari 100K and Craftium, both widely considered challenging benchmarks in sample-efficient RL (keep in mind that the agent is limited to 100K interaction steps, equivalent to roughly 2 hours of gameplay). Our agent achieves **superhuman** performance on all 4 Atari games, demonstrating human-level competitiveness in nontrivial domains.
>
>
> ---
>
>
> `W4`:
> > There is an extra quotation mark at the end of line 446.
>
>  We thank reviewer 8WiC for raising this issue, we have fixed it, and it will be reflected in the next revision of the paper.
>
>
> ---
>
> [1] Hafner, Danijar, et al. "Mastering atari with discrete world models." arXiv preprint arXiv:2010.02193 (2020).
>
> [2] Alonso, Eloi, et al. "Diffusion for world modeling: Visual details matter in atari." Advances in Neural Information Processing Systems 37 (2024): 58757-58791.
>
> [3] Micheli, Vincent, et al. “Transformers Are Sample-Efficient World Models.” The Eleventh International Conference on Learning Representations , 2023, https://openreview.net/forum?id=vhFu1Acb0xb.
>
> [4] Cohen, Lior, et al. "Uncovering Untapped Potential in Sample-Efficient World Model Agents." arXiv preprint arXiv:2502.11537 (2025).

---

> ### Author Response · Authors · 2025-11-19
>
> #### Questions
>
> `Q1`:
> > Are there any possible exploratory directions for extending the idea of Horizon Imagination to continuous action spaces?
>
> `A1`: Yes. While the Horizon schedule is very general, supporting continuous action spaces requires adapting the stable action sampling mechanism $a(\pi, \omega)$.
>
> One immediate way to extend $a(\pi, \omega)$ to continuous spaces could be to sample a $\omega \sim \mathcal{N}(\mathbf{0}, \mathbf{I})$ and define $a(\pi, \omega) = \mu_{\pi} + \omega \sigma_{\pi}$, as in the reparameterization trick, for a Gaussian policy that outputs a mean $\mu_{\pi}$ and standard deviation $\sigma_{\pi}$.
>
> This would minimize action changes when $\mu_{\pi}, \sigma_{\pi}$ remains fixed, reduce action difference otherwise, and guarantee that $a(\pi, \cdot)$ follows the policy distribution.
>
> Potentially, this could spare the computational overhead involved in classifier guidance based methods (prior approaches).
>
> ---
>
> `Q2`:
> > Algorithm 1 only includes the world model ...
>
> `A2`:
> We agree that a complete pseudocode of the entire training pipeline would improve clarity.
> Such pseudocode will be included in the next revision of the paper.
>
> We thank reviewer 8WiC for this valuable suggestion.

---

### Official Review · Reviewer_irHg · 2025-10-30

**Soundness:** 2
**Presentation:** 2
**Contribution:** 3
**Rating:** 4
**Confidence:** 2

**Summary:**

This work proposes Horizon Imagination that tackles a common problem in model-based reinforcement learning with diffusion world models. In each step, the policy receives an observation from the world model, which is subsequently used to generate an action. The world model then predicts the next observation. This sequential dependency on the policy leads to a long inference time that is a bottleneck in fast training. Horizon Imagination, therefore, proposes denoising multiple future observations in parallel in conjunction with a stabilization mechanism to account for the changes in the actions and a sampling schedule. The effectiveness of the proposed method is shown on several benchmark environments.

**Strengths:**

- The paper tackles an interesting and important problem in world models for policy learning.  I think the proposed method is quite important for the RL field.

**Weaknesses:**

- I think the presentation of the paper can be improved.

 - Please also see my questions.

**Questions:**

- I found it a bit difficult to follow. As a non-expert in Flows/Diffusion models, I was a bit confused with the notation. Is it for every z several denoising time steps sampled in Eq. 1, or is it only one denoising time step per sample?

- Section 4.2 states that Eq. 1 requires knowing all actions; however, the velocity field v_\theta is expecting actions $a_{<t}$, which are all actions in the history starting from the current time-step $t$. Doesn't this mean that there is actually no need for knowing future actions, as stated in Section 4.2?

- From my understanding, the paper proposes a method that turns the strict autoregressive structure of the world model (i.e., policy gives action, world model predicts observation, ...) into a parallel inference version. However, I don't understand how this autoregressive structure is still retained in the current version? I think the autoregressive prediction structure still needs to hold in general for world models.

- Given that the policy is queried for noisy variables z, aren't there superscripts to the actions in Eq. (1) missing?

- Section 4.3 mentions using the REINFORCE algorithm. It is well-known that the REINFORCE has high variance in the gradient estimates, whereas the reparameterization trick has smaller variance in the gradients. Is there a reason why the reparameterization trick was not used, e.g., in connection with the Gumbel softmax [1] policy representation that allows using the reparameterization trick?

- Is the proposed method also applicable to continuous state-action spaces? What would be necessary in this case?

[1] E. Jang, et al. Categorical Reparameterization with Gumbel-Softmax. ICLR 2017.

---

> ### Author Response · Authors · 2025-11-19
>
> We thank reviewer irHg for their time and effort in reviewing our paper and for their constructive and valuable feedback.
>
> We appreciate the reviewer’s positive assessment regarding the importance of the problem and the proposed method. We would also like to note the rigor and methodological soundness of our empirical study, which features extensive evaluations and ablations across diverse benchmarks and investigates several important aspects of the proposed method.
> We hope the reviewer may consider this as an additional strength of the submission.
>
> We are currently revising the paper to incorporate the changes discussed below. The updated version will be available before the end of the discussion phase, and we will issue a notification when it is posted.
>
> #### Weaknesses
> `W1`:
> > I think the presentation of the paper can be improved.
>
> We have addressed the presentation issues raised in reviewer irHg’s questions (see details below). If any further issues remain, we would be happy to clarify them.
>
>
> #### Questions
> `Q1`:
> > I found it a bit difficult to follow. As a non-expert ...
>
>
> `A1`:
> In the context of Eq. (1), a single denoising time $\tau_t$ is sampled for every observation $z_t$.
>
> The *training* of diffusion models is often done as follows:
> 1. Sample a data batch $x$
> 2. Sample a corresponding noise batch $\epsilon$
> 3. Sample denoising times $\tau$ (in our case, one time value $\tau_t$ per observation $z_t$, thus $h$ values per sequence)
> 4. Compute the noised observations sequence $x^\tau$ (model input)
> 5. Compute the model outputs $\hat{y} = f_{\theta}(x^\tau, \tau)$
> 6. Optimize the output $\hat{y}$ against a regression target $y$ which typically depends on $\epsilon$ and / or the clean data $x$.
>
> Equation (1) corresponds to steps 5-6, where each term in the summation refers to a specific observation/time step in the sequence. Accordingly, for every observation $z_t$ we sample one denoising time $\tau_t$. All terms in the summation are computed **in parallel** within a single forward pass.
>
> Notably, Eq. 1 emphasizes the causal relationship between inputs and outputs, where each output element depends only on current and past inputs, never on future ones.
>
> We thank reviewer irHg for raising this point. We will clarify this in the revision and include pseudo code to make the procedure easier to follow.
>
>
> `Q2`:
> > Section 4.2 states that Eq. 1 requires ...
>
> `A2`: Not exactly. While it is true that $v_{\theta}$ is causal and can be used autoregressively to avoid knowing multiple actions in advance, our goal is to generate (denoise) multiple steps in parallel. For that purpose, all future actions associated with those steps must be available in advance.
>
>
>
> ---
>
>
> `Q3`:
> > From my understanding, the paper ...
>
>
> `A3`: The parallel generation maintains a *causal* structure through the causal Attention mask, ensuring that each token can only attend to tokens from its own timestep or earlier ones, preventing any access to future information.
> Furthermore, the gradual denoising process enables the model to adapt to fluctuations in actions or dynamics while still respecting this causal structure.
> Lastly, the decay horizon $\nu$ of the Horizon schedule enables precise control over the amount of frames denoised in parallel.
>
> A priori, it was not obvious that such a parallel approach would succeed in practice. However, our experiments consistently demonstrate its empirical viability.
>
>
> ---
>
>
> `Q4`:
> > Given that the policy is queried for noisy variables z, aren't there superscripts to the actions in Eq. (1) missing?
>
>
> `A4`: No, Eq. (1) refers to the world model training stage. At train time, there is no multi-step denoising process. Instead, model outputs are computed once per sampled trajectory segment, and the optimization objective is computed following Eq. (1).
>
> In addition, the policy is not used during the world model training stage. There, the actions and observations are pre-collected trajectory segments loaded from the replay buffer.
>
> Please refer to `A1` above for a description of the steps performed during world model training. We will add pseudocode in the next revision of the paper to further clarify this point.

---

> ### Author Response · Authors · 2025-11-19
>
> `Q5`:
> > Section 4.3 mentions using the REINFORCE ...
>
> `A5`: Since improving control performance is not the focus of our paper, we adopted a standard actor-critic algorithm and architecture used by several prominent prior works in the world model agents literature under the sample efficiency setting [1][2][3][4].
>
>
>
> [1] Hafner, Danijar, et al. "Mastering atari with discrete world models." arXiv preprint arXiv:2010.02193 (2020).
>
> [2] Alonso, Eloi, et al. "Diffusion for world modeling: Visual details matter in atari." Advances in Neural Information Processing Systems 37 (2024): 58757-58791.
>
> [3] Micheli, Vincent, et al. “Transformers Are Sample-Efficient World Models.” The Eleventh International Conference on Learning Representations , 2023, https://openreview.net/forum?id=vhFu1Acb0xb.
>
> [4] Cohen, Lior, et al. "Uncovering Untapped Potential in Sample-Efficient World Model Agents." arXiv preprint arXiv:2502.11537 (2025).
>
>
> ---
>
>
>
> `Q6`:
> > Is the proposed method also applicable to continuous state-action spaces? What would be necessary in this case?
>
>
> `A6`:
> There is a straightforward extension of the stable action sampling mechanism to continuous action spaces, which we leave for future work:
>
> One immediate way to extend $a(\pi, \omega)$ to continuous spaces could be to sample a $\omega \sim \mathcal{N}(\mathbf{0}, \mathbf{I})$ and define $a(\pi, \omega) = \mu_{\pi} + \omega \sigma_{\pi}$, as in the reparameterization trick, for a Gaussian policy that outputs a mean $\mu_{\pi}$ and standard deviation $\sigma_{\pi}$.
>
> This would minimize action changes when $\mu_{\pi}, \sigma_{\pi}$ remains fixed, reduce action difference otherwise, and guarantee that $a(\pi, \cdot)$ follows the policy distribution.
>
> Potentially, this could spare the computational overhead involved in classifier guidance based methods (prior approaches).
> We propose to include the above in a future directions discussion.
>
>
> That said, we focus on the discrete case because (i) multiple solutions already exist for continuous actions but none for discrete actions, (ii) typical applications use either discrete or continuous actions but not both simultaneously.
>
> It's also worth noting that while the proposed action sampling method is tailored to the discrete case, our Horizon schedule is very general and can suite a wide range of applications. Essentially, every diffusion / flow method for sequence generation can utilize it.

---

### Official Review · Reviewer_kptM · 2025-11-01

**Soundness:** 3
**Presentation:** 3
**Contribution:** 3
**Rating:** 6
**Confidence:** 2

**Summary:**

The paper proposes Horizon Imagination, an on-policy training procedure for reinforcement learning with diffusion world models that denoise multiple future observations in parallel. This is done by introducing a stable discrete-action sampling mechanism to avoid spurious action flips during denoising, and a novel Horizon schedule that decouples the denoising budget from the decay horizon ν, enabling sub-frame budgets and finer control of compute–quality trade-offs. Experiments on Atari 100K and Craftium show the method maintains control performance with only half the denoising budget and improves generation quality under parallel configurations.

**Strengths:**

The proposed horizon schedule is a neat design: by fixing ν while varying B, it breaks the tight coupling seen in pyramidal schedules, allowing consistent temporal denoising behavior across budgets and enabling sub-frame B < h operation.

The stable action sampling a(π,ω) for discrete policies is elegant and theoretically justified: action changes between denoising steps are bounded below by total variation distance and above by a derived l_1  term, greatly reducing unnecessary flips during denoising.

Solid empirical analysis: (a) action-consistency experiments demonstrate near-optimal behavior vs. TV lower bound and strong improvements vs. naïve sampling; (b) control performance comparison across ν/B settings; (c) generation quality vs. ν/B via FVD and MSE Clarity.

**Weaknesses:**

The proposed method, especially the action sampling, is only for discrete action spaces. This limits applicability to many continuous-control tasks

The paper would benefit if a per-stage runtime analysis and real-time control throughput (fps) comparison were presented to show the improvement.

It would help to connect the theoretical bound to returns—e.g., does reduced action-flip rate correlate with improved advantage estimates or policy gradient variance?

**Questions:**

Do you foresee a principled extension of a(π,ω)to continuous spaces (beyond discretization)?

How sensitive are returns to policy entropy regularization when using a(π,ω)? Is there a sweet spot where action stability helps most?

FVD/MSE trends favor certain ν/B regimes; can you relate those trends directly to control performance (e.g., correlation analyses across seeds)?

---

> ### Author Response · Authors · 2025-11-19
>
> We thank reviewer kptM for their time and effort in reviewing our paper, for their constructive and valuable feedback, and for the positive review.
>
> We appreciate the reviewer’s recognition of the strengths of our work, including the benefits of the Horizon schedule, the theoretically grounded stable action-sampling mechanism, and the thorough, high-standard empirical study.
>
> We are currently revising the paper to incorporate the changes discussed below. The updated version will be available before the end of the discussion phase, and we will issue a notification when it is posted.
>
> #### Weaknesses
> `W1`:
> Only the stable action-sampling mechanism is specific to the discrete setting, while the Horizon schedule is fully general and applies to any diffusion or flow based sequence model. We focus on the discrete case because (i) multiple solutions already exist for continuous actions but none for discrete actions, (ii) typical applications use either discrete or continuous actions but not both simultaneously. Finally, as discussed in `A1` below, our sampling method admits a straightforward extension to the continuous setting, which we will include as a future direction.
>
> ---
>
> `W2`:
> We agree that a detailed runtime analysis and imagination throughput would provide a fuller performance profile. We will include these results in the appendix of the next revision of the paper.
>
> We summarize the runtime analysis below (all values in milliseconds, measured on an RTX 4090):
>
> | Stage (Time in ms) | $B = 16$ | $B = 32$ |
> | :--- | :--- | :--- |
> | **Tokenizer Training Step** | 72.5 | 72.5 |
> | **World Model Training Step** | 133.2 | 133.2 |
> | **Controller Training Step (Total)** | **1071** | **1871** |
> | └─ **Imagination (Total)** | 673 | 1326.5 |
> | &nbsp;&nbsp;&nbsp;&nbsp;└─ Reward Prediction | 5.18 | 5.18 |
> | &nbsp;&nbsp;&nbsp;&nbsp;└─ **Denoising Sum ($B \times$ steps)** | **652.8** | **1305.6** |
> | &nbsp;&nbsp;&nbsp;&nbsp;&nbsp;&nbsp;&nbsp;&nbsp;└─ *Single Step Duration (Avg)* | *40.8* | *40.8* |
> | &nbsp;&nbsp;&nbsp;&nbsp;&nbsp;&nbsp;&nbsp;&nbsp;&nbsp;&nbsp;&nbsp;&nbsp;└─ Action Computation | 6.5 | 6.5 |
> | &nbsp;&nbsp;&nbsp;&nbsp;&nbsp;&nbsp;&nbsp;&nbsp;&nbsp;&nbsp;&nbsp;&nbsp;└─ Denoiser Forward | 34.3 | 34.3 |
>
> ---
>
> `W3`:
> We appreciate this suggestion. We agree that demonstrating the impact of the proposed stable action sampling method on control performance would offer further support for its necessity.
>
> In the revised paper, we add an empirical comparison of control performance (returns) between the stable and naïve action-sampling methods under the $(\nu=4, B=16)$ configuration.
>
> We are currently running the experiments with the naïve baseline. Preliminary results already show a substantial drop in performance when using naïve sampling, highlighting the practical importance of our stable action-sampling mechanism.
>
> We thank reviewer kptM for this valuable recommendation.

---

> ### Author Response · Authors · 2025-11-19
>
> #### Questions
> `A1`: One immediate way to extend $a(\pi, \omega)$ to continuous spaces could be to sample a $\omega \sim \mathcal{N}(\mathbf{0}, \mathbf{I})$ and define $a(\pi, \omega) = \mu_{\pi} + \omega \sigma_{\pi}$, as in the reparameterization trick, for a Gaussian policy that outputs a mean $\mu_{\pi}$ and standard deviation $\sigma_{\pi}$.
>
> This would minimize action changes when $\mu_{\pi}, \sigma_{\pi}$ remains fixed, reduce action difference otherwise, and guarantee that $a(\pi, \cdot)$ follows the policy distribution.
>
> Potentially, this could spare the computational overhead involved in classifier guidance based methods (prior approaches).
>
> ----
>
> `A2`: The entropy regularization coefficient (denoted as "Entropy weight" in Table 5 in the appendix) was fixed throughout our experiments following a standard value used in several prior works [1][2][3][4]. Since we have not explored tuning or changing this value, we believe it is indicative of the robustness of our approach to this hyperparameter.
>
> That said, policy entropy can vary significantly between states / observation sequences and throughout the training. Studying this question in real environments (rather than the controlled setting in Section 5.1) is thus very challenging.
>
> [1] Hafner, Danijar, et al. "Mastering atari with discrete world models." arXiv preprint arXiv:2010.02193 (2020).
>
> [2] Alonso, Eloi, et al. "Diffusion for world modeling: Visual details matter in atari." Advances in Neural Information Processing Systems 37 (2024): 58757-58791.
>
> [3] Micheli, Vincent, et al. “Transformers Are Sample-Efficient World Models.” The Eleventh International Conference on Learning Representations , 2023, https://openreview.net/forum?id=vhFu1Acb0xb.
>
> [4] Cohen, Lior, et al. "Uncovering Untapped Potential in Sample-Efficient World Model Agents." arXiv preprint arXiv:2502.11537 (2025).
>
> ---
>
> `A3`: Relating the FVD/MSE metrics to control performance would require training agents across all $(\nu, B)$ combinations in the online control setting of Section 5.2. We estimate the total runtime of such an experiment to be 74,988 A100 GPU-hours ($\approx$ 8.56 years) at an estimated cost of $\approx$ \\$$96,000$ (see calculation below), which is far beyond our available budget.
>
> That said, we selected the $(\nu=4, B=16)$ and $(\nu=4, B=32)$ configurations in Section 5.2 based on the following considerations:
> 1. Strong FVD/MSE performance, indicating reliable generative quality.
> 2. A conservative choice of $\nu$, since larger values deviate further from the autoregressive baseline and thus pose greater risk.
> 3. Lower computational cost.
> 4. Coverage of both a sub-frame budget and a standard budget, demonstrating performance in both settings.
>
> **Cost Estimation Details:**
> Runtime differences across $\nu$/$B$ configurations arise solely from variation in imagination time, i.e., the policy-training-in-imagination stage. We assume that the imagination time is proportional to the budget $B$, i.e., a 2x increase in $B$ translates to a 2x increase in imagination time.
>
> Hence, given the measured run times of 19 and 27 hours on A100 GPUs for Atari (for $B=16$ and $B=32$, respectively), we estimate a fixed runtime of 11 hours for all other stages, and 8h for the imagination stage under $B=16$. For Craftium, the fixed runtime is estimated to be around 20h due to slower interactions (based on our data).
>
> This results in the following estimated single run runtimes (in A100 hours):
>
> |     $B$  | $8$ | $16$ | $32$ | $64$ | $128$ | $256$ |
> | :-------- | :---: | :----: | :----: | :----: | :----: | :----: |
> | Atari    |  15 |  19  |  27  |  43  | 75   | 139  |
> | Craftium |  24 | 28   | 36   |  52  | 84   | 148  |
>
>
> For Atari, the total number of runs is $6 \times 4 \times 5 = 120$, corresponding to 6 $\nu$ values, 4 games, and 5 seeds.
>
> For Craftium, since the SmallRoom-v0 environment only runs for 30K steps, the total number of runs is $6 \times 3.3 \times 5 = 99$.
>
> The total runtime estimate is given by $120 \times (15+19+\cdots +139) + 99\times(24+28+\cdots + 148)=38160+36828=74988$ A100 hours.
>
> The [cost of A100/hour](https://lambda.ai/pricing) is currently \\$1.29, resulting in a total cost of \\$96,734.52.

---

### Author Response · Authors · 2025-11-24

We thank all reviewers for their constructive and highly valuable feedback.

In light of the reviewers’ comments, we thoroughly revised the relevant sections of both the main paper and the appendix. Key changes include:

1. **Added a new ablation** comparing control performance under the proposed stable action sampling method versus the naive baseline (Section 5.2.3), showing that naive sampling significantly degrades performance (reviewer kptM).
2. **Added a detailed runtime analysis** and imagination-throughput comparison (Appendix B; reviewer kptM).
3. **Included a complete pseudocode** of the entire algorithm, detailing the main stages and key procedures (e.g., world-model training) (Appendix C; reviewers 8WiC, irHg).
4. **Improved the presentation** of Sections 4.1 and 4.2 through a major revision (reviewer irHg).
5. **Added a discussion** on extending the method to continuous action spaces in future work (reviewers kptM, irHg, 8WiC).
6. Fixed a typo (reviewer 8WiC)

Taken together, these revisions lead to a notably stronger and clearer manuscript that we believe addresses the reviewers’ key concerns. We remain ready to clarify any outstanding points.

---

### Meta-Review · Area_Chair_vaiR · 2026-01-07

**Summary:**

Across the three reviews, the submission is viewed as a meaningful efficiency contribution to diffusion/flow-based world-model RL: it parallelizes multi-step imagination and introduces (i) a Horizon schedule that decouples denoising budget from the decay horizon, and (ii) a stable discrete action-sampling mechanism that reduces spurious action flips during denoising while preserving the intended sampling distribution. The main concerns affecting my decision are:

• Generality / scope: the key stabilization mechanism is discrete-action specific, limiting applicability to continuous-control settings.

• Evaluation framing: control results are largely an internal ablation (parallel vs. autoregressive within their agent) without end-to-end comparisons to other established world-model agents, making it harder to contextualize impact.

• Clarity and correctness of exposition: one reviewer found the method hard to follow and raised notation/causality questions, which also affects confidence.

• Systems evidence: reviewers asked for runtime/throughput and stage-wise costs to support the efficiency claim.

**Reviewer Concerns:**

Mostly Addressed:

• Runtime / stage-wise profiling: Authors provided a detailed breakdown (ms per stage; imagination/denoising/action computation) and committed to include it in the appendix.

• Clarity on Eq. (1) sampling and training procedure: Authors clarified that one denoising time is sampled per observation, explained the standard diffusion training pipeline, and promised pseudocode—this resolves the core confusion about “how many denoising steps per sample.”

• Why parallel denoising needs future actions / causality: Response distinguishes causal model structure vs. parallel generation requirement and explains that parallel denoising requires having those actions available; causal masking preserves no-leakage.

• Need for full pipeline pseudocode: Authors agreed and committed to add complete pseudocode—good faith and likely sufficient.

• “Action stability matters for control”: Authors agreed and are adding a direct stable vs. naïve control comparison under the parallel configuration (preliminary results suggest a meaningful drop for naïve sampling).

Partially addressed:

• Discrete-only limitation: The rebuttal proposes a plausible continuous extension via reparameterization with fixed noise for Gaussian policies, but it remains future work (no experiments, no formal treatment). Applicability to continuous-control is still not demonstrated.

• Lack of external end-to-end baselines: Authors argue that absolute performance is irrelevant since the claim is efficiency and there are no direct baselines for parallel discrete diffusion imagination. That argument is reasonable, but from a reader’s perspective, contextual benchmarks (even a limited comparison on compute/return trade-offs against a few known agents) would strengthen positioning.

• Linking theory (flip bounds) to RL learning dynamics: The rebuttal adds a control comparison, but does not yet show deeper linkage (e.g., advantage variance, gradient variance, correlation with stability metrics).

• Presentation quality more broadly: The answers fix several key confusions; however, the reviewer’s overall “hard to follow” impression suggests the revision must materially improve exposition (notation, algorithm boxes, and stage separation).

**Reviewer Scores:**

kptM: 6: no change.

irHg: +1

8WiC: no change.

---

### Decision · Program_Chairs · 2026-01-26

Accept (Poster)